# GENERALIZED SCHRÖDINGER BRIDGE MATCHING

**Guan-Horng Liu**[1*], **Yaron Lipman**[2,3], **Maximilian Nickel**[3], **Brian Karrer**[3],
**Evangelos A. Theodorou**[1], **Ricky T. Q. Chen**[3]
[1]Georgia Institute of Technology  [2]Weizmann Institute of Science  [3]FAIR, Meta
{ghliu, evangelos.theodorou}@gatech.edu
{ylipman, maxn, briankarrer, rtqichen}@meta.com

## ABSTRACT

Modern distribution matching algorithms for training diffusion or flow models directly *prescribe* the time evolution of the marginal distributions between two boundary distributions. In this work, we consider a generalized distribution matching setup, where these marginals are only *implicitly* described as a solution to some task-specific objective function. The problem setup, known as the Generalized Schrödinger Bridge (GSB), appears prevalently in many scientific areas both within and without machine learning. We propose **Generalized Schrödinger Bridge Matching (GSBM)**, a new matching algorithm inspired by recent advances, generalizing them beyond kinetic energy minimization and to account for task-specific state costs. We show that such a generalization can be cast as solving *conditional* stochastic optimal control, for which efficient variational approximations can be used, and further debiased with the aid of path integral theory. Compared to prior methods for solving GSB problems, our GSBM algorithm better preserves a feasible transport map between the boundary distributions throughout training, thereby enabling stable convergence and significantly improved scalability. We empirically validate our claims on an extensive suite of experimental setups, including crowd navigation, opinion depolarization, LiDAR manifolds, and image domain transfer. Our work brings new algorithmic opportunities for training diffusion models enhanced with task-specific optimality structures.

## 1 INTRODUCTION

The distribution matching problem—of learning transport maps that match specific distributions—is a ubiquitous problem setup that appears in many areas of machine learning, with extensive applications in optimal transport (Peyré & Cuturi, 2017), domain adaptation (Wang & Deng, 2018), and generative modeling (Sohl-Dickstein et al., 2015; Chen et al., 2018). The tasks are often cast as learning mappings, $X_0 \mapsto X_1$, such that $X_0 \sim \mu$ $X_1 \sim \nu$ follow the (unknown) laws of two distributions $\mu, \nu$. For instance, diffusion models[1] construct the mapping as the solution to a stochastic differential equation (SDE) whose drift $u_t^\theta(X_t) : \mathbb{R}^d \times [0,1] \to \mathbb{R}^d$ is parameterized by $\theta$:

$$\mathrm{d}X_t = u_t^\theta(X_t)\mathrm{d}t + \sigma\mathrm{d}W_t, \quad X_0 \sim \mu, X_1 \sim \nu. \tag{1}$$

The marginal density $p_t$ induced by (1) evolves as the Fokker Plank equation (FPE; Risken (1996)),

$$\frac{\partial}{\partial t}p_t(X_t) = -\nabla \cdot (u_t^\theta(X_t)\,p_t(X_t)) + \frac{\sigma^2}{2}\Delta p_t(X_t), \quad p_0 = \mu, p_1 = \nu, \tag{2}$$

and prescribing $p_t$ with fixed $\sigma$ uniquely determines a family of parametric SDEs in (1). Indeed, modern successes of diffusion models in synthesizing high-fidelity data (Song et al., 2021; Dhariwal & Nichol, 2021) are attributed, partly, to constructing $p_t$ as a mixture of tractable conditional probability paths (Liu et al., 2023b; Albergo et al., 2023; Tong et al., 2023). These tractabilities enable scalable algorithms that "*match $u_t^\theta$ given $p_t$*", hereafter referred to as *matching algorithms*.

Alternatively, one may also specify $p_t$ implicitly as the optimal solution to some objective function, with examples such as optimal transport (OT; Villani et al. (2009)), or the Schrödinger Bridge

---

*Work done in part as a research intern at FAIR, Meta.

[1]We adopt constant $\sigma \in \mathbb{R}$ throughout the paper but note that all analysis generalize to time-dependent $\sigma_t$.

problem (SB; Schrödinger (1931); Fortet (1940); De Bortoli et al. (2021)). SB generalizes standard diffusion models to arbitrary $\mu$ and $\nu$ with fully nonlinear stochastic processes and, among all possible SDEs that match between $\mu$ and $\nu$, seeks the *unique* $u_t$ that minimizes the kinetic energy.

While finding the transport with minimal kinetic energy can be motivated from statistical physics or entropic optimal transport (Peyré & Cuturi, 2019; Vargas et al., 2021), with kinetic energy often being correlated with sampling efficiency in generative modeling (Chen et al., 2022; Shaul et al., 2023), it nevertheless limits the flexibility in the design of "optimality". Indeed, kinetic energy corresponds to the squared-Euclidean cost in OT, which, despite its popularity, is merely one among numerous alternatives available for deployment (Di Marino et al., 2017). On one hand, it remains debatable whether the $\ell_2$ distance defined in the original data space (*e.g.,* pixel space for images), as opposed to other metrics, are best suited for quantifying the optimality of transport maps. On the other hand, distribution matching in general scientific domains, such as population modeling (Ruthotto et al., 2020), robot navigation (Liu et al., 2018), or molecule simulation (Noé et al., 2020), often involves more complex optimality conditions that require more general algorithms to handle.

To this end, we advocate a generalized setup for distribution matching, previously introduced as the Generalized Schrödinger Bridge problem (GSB; Chen et al. (2015); Chen (2023); Liu et al. (2022)):

$$\min_\theta \int_0^1 \mathbb{E}_{p_t} \left[ \frac{1}{2} \|u_t^\theta(X_t)\|^2 + V_t(X_t) \right] \mathrm{d}t \quad \text{subject to (1) or, equivalently, (2).} \tag{3}$$

GSB is a distribution matching problem—as it still seeks a diffusion model (1) that transports $\mu$ to $\nu$. Yet, in contrast to standard SB, the objective of GSB involves an additional state cost $V_t$ which affects the solution by quantifying a penalty—or equivalently a reward—similar to general decision-making problems. This state cost can also include distributional properties of the random variable $X_t$. Examples of $V_t$ include, *e.g.,* the mean-field interaction in opinion propagation (Gaitonde et al., 2021), quantum potential (Philippidis et al., 1979), or a geometric prior.

Solving GSB problems involves addressing two distinct aspects: **optimality** (3) and **feasibility** (2). Within the set of feasible solutions that satisfy (2), GSB considers the one with the lowest objective in (3) to be optimal. Therefore, it is essential to develop algorithms that search *within* the feasible set for the optimal solution. Unfortunately, existing methods that approximate solutions to (3) either require relaxing feasibility (Koshizuka & Sato, 2023), or, following the design of Sinkhorn algorithms (Cuturi, 2013), prioritize optimality over feasibility. While an exciting line of new matching algorithms (Peluchetti, 2023; Shi et al., 2023) has been developed for SB, *i.e.,* when $V_t := 0$, to ensure that the solutions are proximity to the feasible set throughout training, it remains unclear whether, or how, these "*SB Matching*" (SBM) algorithms can be extended to handle nontrivial $V_t$.

We propose **Generalized Schrödinger Bridge Matching (GSBM)**, a new matching algorithm that generalizes SBM to nontrivial $V_t$. We discuss how such a generalization can be tied to a *conditional stochastic optimal control* (CondSOC) problem, from which existing SBM algorithms can be derived as special cases. We develop scalable solvers to the CondSOC using Gaussian path approximation, further debiased with path integral theory (Kappen, 2005). GSBM inherits a similar algorithmic characterization to its ancestors (Liu et al., 2023b; Shi et al., 2023), in that, during the optimization process, the learned $p_0^\theta$ and $p_1^\theta$ induced by the subsequent solutions of $u_t^\theta$ remain close to the boundary marginals $(\mu, \nu)$ and preserve them exactly under stricter theoretical conditions; see Sec. 3.1 for details. This distinguishes GSBM from prior methods (*e.g.,* Liu et al. (2022)) that learn approximate solutions to the same problem (3) but whose subsequent solutions only approach $(\mu, \nu)$ after final convergence. This further results in a framework that relies solely on samples from $\mu, \nu$—without knowing their densities—and enjoys stable convergence, making it suitable for high dimensional applications. A note on the connection to stochastic optimal control and related works (Liu et al., 2022) can be found in Appendix A. Summarizing, we present the following contributions:

- We propose GSBM, a new matching algorithm for learning diffusion models between two distributions that also respect some task-specific optimality structures in (3) via specifying $V_t$.

- GSBM casts recent matching methods as conditional stochastic optimal control problems, from which nontrivial $V_t$ can be incorporated and solved at scale via Gaussian path approximation.

- Compared to prior methods, *e.g.,* Liu et al. (2022), that also provide approximate solutions to (3), GSBM enjoys stable convergence, improved scalability, and, crucially, maintains a transport map that remains *much* closer to the distribution boundaries throughout the entire training.

| **Algorithm 1** `match` (implicit) | **Algorithm 2** `match` (explicit) |
|---|---|
| **Require:** $p_t$ s.t. $p_0 = \mu$, $p_1 = \nu$ | **Require:** $p_t := \mathbb{E}_{p_{0,1}}[p_{t\|0,1}]$ s.t. $p_0{=}\mu$, $p_1{=}\nu$, $u_{t\|0,1}$ |
|   **repeat** |   **repeat** |
|     Sample $X_0 \sim p_0$, $X_t \sim p_t$, $X_1 \sim p_1$ |     Sample $X_0, X_1 \sim p_{0,1}$, $X_t \sim p_{t\|0,1}$ |
| |     Compute $u_{t\|0,1}$ given $(X_0, X_t, X_1)$ |
|     Take gradient step w.r.t. $\mathcal{L}_{\text{implicit}}(\theta)$ |     Take gradient step w.r.t. $\mathcal{L}_{\text{explicit}}(\theta)$ |
|   **until** converges |   **until** converges |
|   **return** $\nabla s_t^{\theta^\star}$ |   **return** $u_t^{\theta^\star}$ |

- Through extensive experiments, we showcase GSBM's remarkable capabilities across a variety of distribution matching problems, ranging from standard crowd navigation and 3D navigation over LiDAR manifolds, to high-dimensional opinion modeling and unpaired image translation.

## 2    PRELIMINARIES: MATCHING DIFFUSION MODELS GIVEN PROB. PATHS

As mentioned in Sec. 1, the goal of matching algorithms is to learn a SDE parametrized with $u_t^\theta$ such that its FPE (2) matches some prescribed marginal $p_t$ for all $t \in [0, 1]$. In this section, we review two classes of matching algorithms that will play crucial roles in the development of our GSBM.

**Entropic Action Matching (implicit).**    This is a recently proposed matching method (Neklyudov et al., 2023) that learns the unique gradient field governing an FPE prescribed by $p_t$. Specifically, let $s_t^\theta(X_t) : \mathbb{R}^d \times [0, 1] \to \mathbb{R}$ be a parametrized function, then the unique gradient field $\nabla s_t^\theta(X_t)$ —by which the probability density of the pushforward from $\mu$ at time 0 to $t$ matches $p_t$—minimizes

$$\mathcal{L}_{\text{implicit}}(\theta) := \mathbb{E}_\mu \left[ s_0^\theta(X_0) \right] - \mathbb{E}_\nu \left[ s_1^\theta(X_1) \right] + \int_0^1 \mathbb{E}_{p_t} \left[ \frac{\partial s_t^\theta}{\partial t} + \frac{1}{2} \|\nabla s_t^\theta\|^2 + \frac{\sigma^2}{2} \Delta s_t^\theta \right] \mathrm{d}t. \quad (4)$$

Neklyudov et al. (2023) showed that $\mathcal{L}_{\text{implicit}}(\theta)$ *implicitly matches* a unique gradient field $\nabla s_t^\star$ with least kinetic energy, *i.e.,* it is equivalent to minimizing $\int_0^1 \mathbb{E}_{p_t} \left[ \frac{1}{2} \|\nabla s_t^\theta - \nabla s_t^\star\|^2 \right] \mathrm{d}t$, where

$$\nabla s_t^\star = \arg\min_{u_t} \int_0^1 \mathbb{E}_{p_t} \left[ \frac{1}{2} \|u_t(X_t)\|^2 \right] \mathrm{d}t \text{ subject to (2).}$$

**Bridge & Flow Matching (explicit).**    If the $p_t$ can be factorized into $p_t := \mathbb{E}_{p_{0,1}}[p_t(X_t|x_0, x_1)]$, where the conditional density $p_t(X_t|x_0, x_1)$—denoted with the shorthand $p_{t|0,1}$—is associated with an SDE, $\mathrm{d}X_t = u_t(X_t|x_0, x_1)\mathrm{d}t + \sigma \mathrm{d}W_t$, then it can be shown (Lipman et al., 2023; Albergo & Vanden-Eijnden, 2023; Liu et al., 2023a) that the minimizer of (see Appendix C.1 for the derivation)

$$\mathcal{L}_{\text{explicit}}(\theta) := \int_0^1 \mathbb{E}_{p_{0,1}} \mathbb{E}_{p_{t|0,1}} \left[ \frac{1}{2} \|u_t^\theta(X_t) - u_t(X_t|x_0, x_1)\|^2 \right] \mathrm{d}t, \quad (5)$$

similar to $\nabla s_t^{\theta^\star}$, also satisfies the FPE prescribed by $p_t$. In other words, $u_t^{\theta^\star}$ preserves $p_t$ for all $t$.

**Implicit *vs.* explicit matching losses.**    While Entropic Action Matching presents a general matching method with the least assumptions, its implicit matching loss $\mathcal{L}_{\text{implicit}}$ (4) scales unfavorably to high-dimensional applications, due to the need to approximate the Laplacian (using *e.g.,* Hutchinson (1989)), and also introduces unquantifiable bias when optimizing over a restricted family of functions such as deep neural networks. The explicit matching loss $\mathcal{L}_{\text{explicit}}$ (5) offers a computationally efficient alternative but requires additional information, namely $u_t(X_t|x_0, x_1) \equiv u_{t|0,1}$. Remarkably, in both cases, the minimizers preserve the prescribed marginal $p_t$. Hence, if $p_0 = \mu$ and $p_1 = \nu$, these matching algorithms shall *always* return a feasible solution—a diffusion model that matches between $\mu$ and $\nu$. We summarize the aforementioned two methods in Alg. 1 and 2.

## 3    GENERALIZED SCHRÖDINGER BRIDGE MATCHING (GSBM)

We propose **Generalized Schrödinger Bridge Matching (GSBM)**, a novel matching algorithm that, in contrast to those in Sec. 2 assuming prescribed $p_t$, concurrently optimizes $p_t$ to minimize (3) subject to the feasibility constraint (2). All proofs can be found in Appendix B.

### 3.1 ALTERNATING OPTIMIZATION SCHEME

Let us revisit the GSB problem (3), particularly its FPE constraint in (2). Recent advances in dynamic optimal transport (Liu et al., 2023b) and SB (Peluchetti, 2023) propose a decomposition of this dynamical constraint into two components: the marginal $p_t$, $t \in (0, 1)$, and the joint coupling between boundaries $p_{0,1}$, and employ alternating optimization between $u_t^\theta$ and $p_t$. Specifically, these optimization methods, which largely inspired recent variants of SB Matching (Shi et al., 2023) and our GSBM, generally obey the following recipe, *alternating* between two stages:

Stage 1: Optimize the objective, in our case, (3) w.r.t. the drift $u_t^\theta$ given fixed $p_{t \in [0,1]}$.

Stage 2: Optimize (3) w.r.t. the marginals $p_{t \in (0,1)}$ given the coupling $p_{0,1}^\theta$ defined by $u_t^\theta$.

Notice particularly that the optimization posed in Stage 1 resembles the matching algorithms in Sec. 2. We make the connection concrete in the following proposition:

**Proposition 1** (Stage 1). *The unique minimizer to Stage 1 coincides with $\nabla s_t^\star(X_t)$.*

This may seem counter-intuitive at first glance, both the kinetic energy and $V_t$ show up in (3). This is due to the fact that $u_t^\theta$ no longer affects the value of $\mathbb{E}_{p_t}[V_t(X_t)]$ once $p_t$ is fixed, and out of all $u_t^\theta$ whose probability density matches $p_t$, the gradient field is the unique minimizer of kinetic energy.[2] We emphasize that the role of Stage 1 is to learn a $u_t^\theta$ given the prescribed $p_t$ and provides a better coupling $p_{0,1}^\theta$ which is then used to refine $p_t$ in Stage 2. Therefore, the explicit matching loss (5) provides just the same algorithmic purpose. Though it only upper-bounds the objective of Stage 1, its solution often converges stably from a measure-theoretic perspective (see Appendix C.1 for more explanations). In practice, we find that $\mathcal{L}_{\text{explicit}}$ performs as effectively as $\mathcal{L}_{\text{implicit}}$, while exhibiting significantly improved scalability, hence better suited for high-dimensional applications.

We now present our main result, which shows that Stage 2 of GSBM can be cast as a variational problem, where $V_t$ appears through optimizing a conditional controlled process.

**Proposition 2** (Stage 2; Conditional stochastic optimal control; CondSOC). *Let the minimizer to Stage 2 be factorized by $p_t(X_t) = \int p_t(X_t|x_0, x_1) p_{0,1}^\theta(x_0, x_1) \mathrm{d}x_0 \mathrm{d}x_1$ where $p_{0,1}^\theta$ is the boundary coupling induced by solving (1) with $u_t^\theta$. Then, $p_t(X_t|x_0, x_1) \equiv p_{t|0,1}$ solves :*

$$\min_{u_{t|0,1}} \mathcal{J} := \int_0^1 \mathbb{E}_{p_t(X_t|x_0,x_1)} \left[ \frac{1}{2} \|u_t(X_t|x_0, x_1)\|^2 + V_t(X_t) \right] \mathrm{d}t \tag{6a}$$

$$s.t.\ \mathrm{d}X_t = u_t(X_t|x_0, x_1)\mathrm{d}t + \sigma \mathrm{d}W_t, \quad X_0 = x_0, \ X_1 = x_1 \tag{6b}$$

Note that (6) differs from (3) only in the boundary conditions, where the original distributions $\mu, \nu$ are replaced by the two end-points $(x_0, x_1)$ drawn from the coupling induced by $u_t^\theta$. Generally, the solution to (6) is not known in closed form, except in special cases. In Lemma 3, we show such a case when $V(x)$ is quadratic and $\sigma > 0$. Note that as $V(x)$ vanishes, $p_{t|0,1}$ in Lemma 3 collapses to the Brownian bridge and GSBM recovers the matching algorithm appearing in prior works (Liu, 2022; Shi et al., 2023) that approximate OT/SB, as summarized in Table 1. This suggests that our Prop. 2 directly generalizes them to nontrivial $V_t$.

Table 1: Solutions to (6) w.r.t. different $V_t$, and how they link to different methods, including Rectified flow (Liu et al., 2023b), DSBM (Shi et al., 2023), and our GSBM.

| Method | $V_t(x)$ | $p_t(X_t|x_0, x_1)$ |
|---|---|---|
| RecFlow ($\sigma=0$) | 0 | straight line (Lemma 3 with $\alpha, \sigma \to 0$) |
| DSBM ($\sigma>0$) | 0 | Brownian bridge (Lemma 3 with $\alpha \to 0$) |
| **GSBM** ($\sigma \geq 0$) | quadratic arbitrary | Lemma 3 Sec. 3.2 |

**Lemma 3** (Analytic solution to (6) for quadratic $V$ and $\sigma > 0$). *Let $V(x) := \alpha \|\sigma x\|^2$, $\alpha, \sigma > 0$, then the optimal solution to (6) follows a Gaussian path $X_t^\star \sim \mathcal{N}(c_t x_0 + e_t x_1, \gamma_t^2 \boldsymbol{I}_d)$, where*

$$c_t = \frac{\sinh(\eta(1-t))}{\sinh \eta}, \quad e_t = \cosh(\eta(1-t)) - c_t \cosh \eta, \quad \gamma_t = \sigma \sqrt{\frac{\sinh(\eta(1-t))}{\eta} e_t}, \quad \eta = \sigma\sqrt{2\alpha}.$$

*These coefficients recover Brownian bridges as $\alpha \to 0$ and, if further $\sigma \to 0$, straight lines.*

---

[2]Notably, the absence of $V_t$ in optimization was previously viewed as an issue for Sinkhorn methods aimed at solving (3) (Liu et al., 2022). Yet, in GSBM, it appears naturally from how the optimization is decomposed.

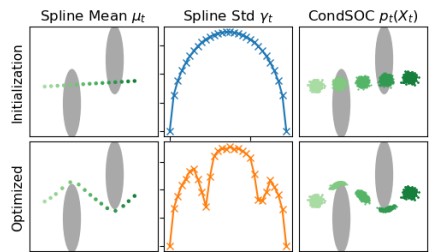

Figure 1: Example of spline optimization (Alg. 3) for $\mu_t \in \mathbb{R}^2, \gamma_t \in \mathbb{R}$, and the resulting CondSOC (6) solution.

---

**Algorithm 3** `SplineOpt`

---

**Require:** $x_0, x_1, \{X_{t_k}\}$ where $0 < t_1 < \cdots < t_K < 1$
    Initialize $\mu_t, \gamma_t$ with (9)
    **for** $m = 0$ **to** $M$ **do**
        Sample $X_t^i \sim \mathcal{N}(\mu_t, \gamma_t^2 \boldsymbol{I}_d)$ with (7)
        Compute $V_t(X_t^i)$ and $u_t(X_t^i | x_0, x_1)$ with (8)
        Estimate the objective $\mathcal{J}$ in (6a)
        Take gradient step w.r.t control pts. $\{X_{t_k}, \gamma_{t_k}\}$
    **end for**
    **return** $p_{t|0,1}$ parametrized by optimized $\mu_t, \gamma_t$

---

### 3.2 APPROXIMATE SOLUTIONS TO CONDSOC FOR GENERAL NONLINEAR STATE COST

We develop methods for searching the approximate solutions to CondSOC in (6), when $V_t$ is neither quadratic nor degenerate. As we need to search for every pair $(x_0, x_1)$ drawn from $p_{0,1}^\theta$, we seek efficient methods that are parallelizable and simulation-free—outside of sampling from $p_{0,1}^\theta$.

**Gaussian path approximation.** Drawing inspirations from Lemma 3, and since the boundary conditions of (6) are simply two fixed points, we propose to approximate the solution to (6) as a Gaussian probability path pinned at $x_0, x_1$:

$$p_t(X_t | x_0, x_1) \approx \mathcal{N}(\mu_t, \gamma_t^2 \boldsymbol{I}_d), \quad \text{s.t.} \quad \mu_0 = x_0, \ \mu_1 = x_1, \ \gamma_0 = \gamma_1 = 0, \tag{7}$$

where $\mu_t \in \mathbb{R}^d$ and $\gamma_t \in \mathbb{R}$ are respectively the time-varying mean and standard deviation. An immediate result following from (7) is a closed-form conditional drift (Särkkä & Solin, 2019) (see Appendix C.2 for the derivation):

$$u_t(X_t | x_0, x_1) = \partial_t \mu_t + a_t(X_t - \mu_t), \quad \text{where } a_t := \frac{1}{\gamma_t}\left( \partial_t \gamma_t - \frac{\sigma^2}{2\gamma_t} \right). \tag{8}$$

Notice that $a_t \in \mathbb{R}$ is a time-varying scalar. Hence, however complex $\mu_t, \sigma_t$ may be, the underlying SDE (with the conditional drift $u_t(X_t | x_0, x_1)$ in (8)) remains linear. Also note that this drift, since it is a gradient field, is the kinetic optimal choice out of all drifts that generate $p_t(X_t | x_0, x_1)$.

**Spline optimization.** To facilitate efficient optimization, we parametrize $\mu_t, \sigma_t$ respectively as $d$- and 1-D splines with some control points $X_{t_k} \in \mathbb{R}^d$ and $\gamma_{t_k} \in \mathbb{R}$ sampled sparsely and uniformly along the time steps $0 < t_1 < ... < t_K < 1$:

$$\mu_t := \text{Spline}(t; \ x_0, \{X_{t_k}\}, x_1) \qquad \gamma_t := \text{Spline}(t; \ \gamma_0 = 0, \{\gamma_{t_k}\}, \gamma_1 = 0). \tag{9}$$

Notice that the parameterization in (9) satisfy the boundary in (7), hence remains as a feasible solution to (6b) however $\{X_{t_k}, \gamma_{t_k}\}$ change. The number of control points $K$ is much smaller than discretization steps ($K \leq 30$ for all experiments). This significantly reduces the memory complexity compared to prior works (Liu et al., 2022), which require caching entire discretized SDEs.

Alg. 3 summarizes the spline optimization, which, crucially, involves no simulation of an SDE (6b). This is because we optimize using just independent samples from $p_{t|0,1}$, which are known in closed form. Since the CondSOC problem relaxes distributional boundary constraints to just two endpoints, our computationally efficient variational approximation holds out very well in practice. Furthermore, since we only need to optimize very few spline parameters, we did not find the need to consider amortized variational inference (Kingma & Welling, 2014). Finally, note that since we solve CondSOC for each pair $(x_0, x_1) \sim p_{0,1}^\theta$ and later marginalize to construct $p_t$, we need not explicitly consider mixture solutions for $p_{t|0,1}$, as long as we sample sufficiently many pairs of $(x_0, x_1)$. In practice, we initialize $\{X_{t_k}\}$ from $u_t^{\theta}$,[3] and $\gamma_{t_k} := \sigma\sqrt{t_k(1 - t_k)}$ from the standard deviation of the Brownian bridge. Figure 1 demonstrates a 2D example.

**Resampling using path integral theory.** In cases where the family of Gaussian probability paths is not sufficient for modeling solutions of (6), a more rigorous approach involves the path integral theory (Kappen, 2005), which provides analytic expression to the optimal density of (6) given any sampling distribution with sufficient support. We discuss this in the following proposition.

---

[3]This induces no computational overhead, as $\{X_{t_k}\}$ are intermediate steps when simulating $x_0, x_1 \sim p_{0,1}^\theta$.

---

**Algorithm 5** Generalized Schrödinger Bridge Matching (GSBM)

---

1: Initialize $u_t^\theta$ and set $p_t$ as the solution to (6) with independent coupling $p_{0,1} := \mu \otimes \nu$
2: **repeat**
3:   $u_t^\theta \leftarrow \text{match}(p_t)$ or $\text{match}(p_t, u_{t|0,1})$                                    ▷ Alg. 1 or 2
4:   Sample $X_0, X_1, X_{t_k}$ from $u_t^\theta$ on timesteps $0 < t_1 < \cdots < t_K < 1$
5:   Optimize $p_{t|0,1}, \mu_t, \gamma_t \leftarrow \text{SplineOpt}(X_0, X_1, \{X_{t_k}\})$                   ▷ Alg. 3
6:   Determine $u_{t|0,1}$ from $\mu_t, \gamma_t$ using (8)
7:   **if** apply path integral resampling **then**                                                    ▷ optional step
8:     $p_{t|0,1} \leftarrow \text{ImptSample}(p_{t|0,1})$                                               ▷ Alg. 4
9:   **end if**
10: **until** converges

---

**Proposition 4** (Path integral solution to (6)). *Let $r(\bar{X}|x_0, x_1)$ be a distribution absolutely contin-uous w.r.t. the Brownian motion, denoted $q(\bar{X}|x_0)$, where we shorthand $\bar{X} \equiv X_{t \in [0,1]}$. Suppose $r$ is associated with an SDE $dX_t = v_t(X_t)dt + \sigma dW_t$, $X_0 = x_0$, $X_1 = x_1$, $\sigma > 0$. Then, the optimal, path-integral solution to (6) can be obtained via*

$$p^\star(\bar{X}|x_0, x_1) = \tfrac{1}{Z}\omega(\bar{X}|x_0, x_1)r(\bar{X}|x_0, x_1), \tag{10}$$

*where $Z$ is the normalization constant and $\omega$ is the importance weight:*

$$\omega(\bar{X}|x_0, x_1) := \exp\left(-\int_0^1 \frac{1}{\sigma^2}\left(V_t(X_t) + \frac{1}{2}\|v_t(X_t)\|^2\right)dt - \int_0^1 \frac{1}{\sigma}v_t(X_t)^\top dW_t\right). \tag{11}$$

In practice, $r(\bar{X}|x_0, x_1)$ can be any distribution, but the closer it is to the optimal $p^\star$, the lower the variance of the importance weights $\omega$ (Kappen & Ruiz, 2016). It is therefore natural to consider us-ing the aforementioned Gaussian probability paths as $r(\bar{X}|x_0, x_1)$, properly optimized with Alg. 3, then resampled following proportional to (10)—this is

---

**Algorithm 4** ImptSample

---

**Require:** $p_{t|0,1}, \mu_t, \gamma_t, \sigma > 0$
  Sample $X_{t \in [0,1]}^i$ from (6b) given (8)
  Compute $\omega^i$ with (11) and $Z = \sum_i \omega^i$
  **return** Resample $p_{t|0,1}$ with (10).

---

equivalent in expectation to performing self-normalized importance sampling but algorithmically simpler and helps reduce variance when many samples have low weight values. While path in-tegral resampling may make $\mathcal{L}_{\text{explicit}}$ less suitable due to the change in the conditional drift gov-erning $p^\star(\bar{X}|x_0, x_1)$ from (8), we still observe empirical, sometimes significant, improvement (see Sec. 4.4). Overall, we propose path integral resampling as an optional step in our GSBM algo-rithm, as it requires sequential simulation.[4] In practice, we also find empirically that the Gaussian probability paths alone perform sufficiently well and is easy to use at scale.

## 3.3 Algorithm outline & convergence analysis

We summarize our GSBM in Alg. 5, which, as previously sketched in Sec. 3.1, alternates between Stage 1 (line 3) and Stage 2 (lines 4-9). In contrast to DSBM (Shi et al., 2023), which constructs analytic $p_{t|0,1}$ and $u_{t|0,1}$ when $V_t = 0$, our GSBM solves the underlying CondSOC (6) with nontriv-ial $V_t$. We stress that these computations are easily parallelizable and admit fast converge due to our efficient parameterization, hence inducing little computational overhead (see Sec. 4.4).

We note that **GSBM (Alg. 5) remains functional even when** $\sigma = 0$, although the GSB problem (3) was originally stated with $\sigma > 0$ (Chen et al., 2015). In such cases, the implicit and explicit matching still preserve $p_t$ and correspond, respectively, to action (Neklyudov et al., 2023) and flow matching (Lipman et al., 2023), and CondSOC corresponds to a generalized geodesic path accounting for $V_t$.

Finally, we provide convergence analysis in the following theorems.

**Theorem 5** (Local convergence). *Let the intermediate result of Stage 1 and 2, after repeating $n$ times, be $\theta^n$. Then, its objective value in (3), $\mathcal{L}(\theta^n)$, is monotonically non-increasing as $n$ grows:*

$$\mathcal{L}(\theta^n) \geq \mathcal{L}(\theta^{n+1}).$$

**Theorem 6.** *The optimal solution to GSB problem (3) is a fixed point of GSBM, Alg. 5.*

---

[4]We note that simulation of trajectories $\bar{X} \equiv X_{t \in [0,1]}$ from (6a,8) can be done efficiently by computing the covariance function, which requires merely solving an 1D ODE; see Appendix C.2 for a detailed explanation.

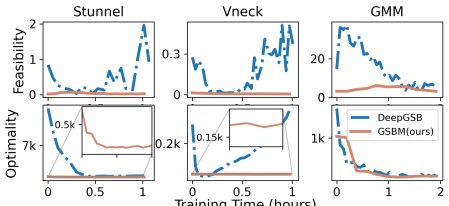

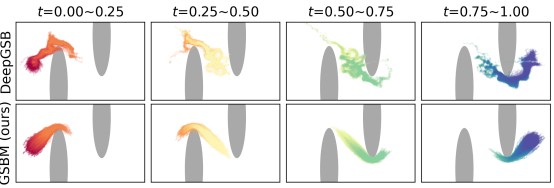

Figure 2: Feasibility *vs.* optimality on three crowd navigation tasks with mean-field cost.

Figure 3: Simulation of SDEs with the $u_t^\theta$ after long training. Notice how DeepGSB diverges drastically from our GSBM, which satisfies feasibility at all time.

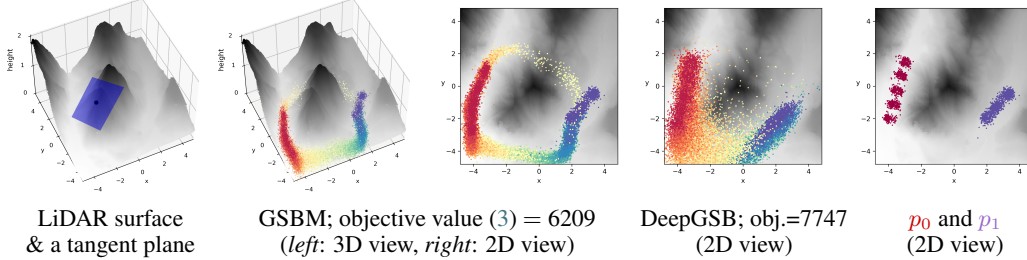

| LiDAR surface & a tangent plane | GSBM; objective value (3) = 6209 (*left*: 3D view, *right*: 2D view) | DeepGSB; obj.=7747 (2D view) | $p_0$ and $p_1$ (2D view) |

Figure 4: Crowd navigation over a LiDAR surface. Height is denoted by the grayscale color.

# 4 EXPERIMENT

We test out GSBM on a variety of distribution matching tasks, each entailing its own state cost $V_t(x)$. By default, we use the explicit matching loss (5) without path integral resampling, mainly due to its scalability, but ablate their relative performances in Sec. 4.4. Our GSBM is compared primarily to DeepGSB (Liu et al., 2022), a Sinkhorn-inspired machine learning method that outperforms existing deep methods (Ruthotto et al., 2020; Lin et al., 2021). Other details are in Appendix D.

## 4.1 CROWD NAVIGATION WITH MEAN-FIELD AND GEOMETRIC STATE COSTS

We first validate our GSBM in solving crowd navigation, a canonical example for the GSB problem (3). Specifically, we consider the following two classes of tasks (see Appendix D.1 for details):

**Mean-field interactions.** These are synthetic dataset in $\mathbb{R}^2$ introduced in DeepGSB, where the state cost $V_t$ consists of two components: an obstacle cost that assesses the physical constraints, and an "mean-field" interaction cost between individual agents and the population density $p_t$:

$$V_t(x) = L_{\text{obstacle}}(x) + L_{\text{interact}}(x; p_t), \quad L_{\text{interact}}(x; p_t) = \begin{cases} \log p_t(x) & \text{(entropy)} \\ \mathbb{E}_{y \sim p_t}\left[\frac{2}{\|x-y\|^2+1}\right] & \text{(congestion)} \end{cases}. \quad (12)$$

Both entropy and congestion costs are fundamental elements of mean-field games. They measure the costs incurred for individuals to stay in densely crowded regions with high population density.

**Geometric surfaces defined by LiDAR.** A more realistic scenario involves navigation through a complex geometric surface. In particular, we consider surfaces observed through LiDAR scans of Mt. Rainier (Legg & Anderson, 2013), thinned to 34,183 points (see Fig. 4). We adopt the state cost

$$V(x) = L_{\text{manifold}}(x) + L_{\text{height}}(x), \quad L_{\text{manifold}}(x) = \|\pi(x) - x\|^2, \quad L_{\text{height}}(x) = \exp\left(\pi^{(z)}(x)\right), \quad (13)$$

where $\pi(x)$ projects $x$ to an approximate tangent plane fitted by a $k$-nearest neighbors (see Fig. 4 and Appendix D.3) and $\pi^{(z)}(x)$ refers to the $z$-coordinate of $\pi(x)$, *i.e.,* its height.

Figure 2 tracks the feasibility and optimality, measured by $\mathcal{W}_2(p_1^\theta, \nu)$ and (3), of three mean-field tasks (Stunnel, Vneck, GMM). On all tasks, our GSBM maintains feasible solutions throughout training while gradually improving optimality. In contrast, due to the lack of convergence analysis, training DeepGSB exhibits relative instability and occasional divergence (see Fig. 3). As for the geometric state costs, Fig. 4 demonstrates how our GSBM faithfully recovers the desired multi-modal distribution: It successfully identify two viable pathways with low state cost, one of which bypasses the saddle point. In constrast, DeepGSB generates only uni-modal distributions with samples scattered over tall mountain regions with high state cost, yielding a higher objective value (3).

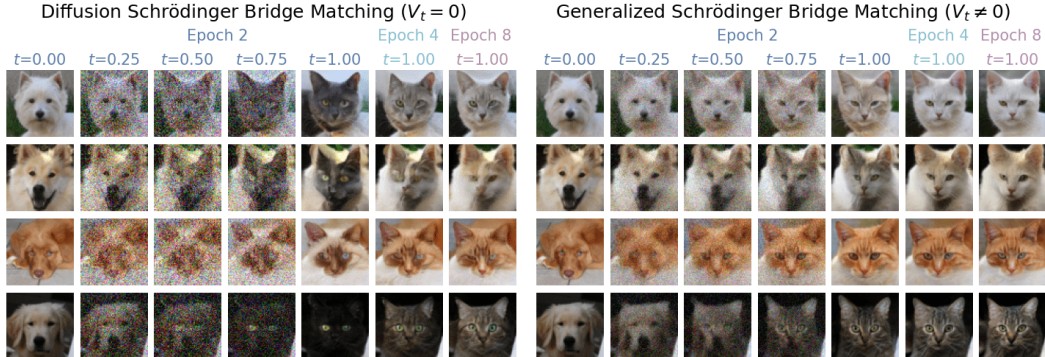

Figure 5: Comparison between DSBM (Shi et al., 2023) and our GSBM on: (*leftmost 5 columns*) the generation processes and (*rightmost 3 columns*) their couplings $p^\theta(X_1|X_0)$ during training. By constructing $V_t$ via a latent space, GSBM exhibits **faster convergence** and **yields better couplings**.

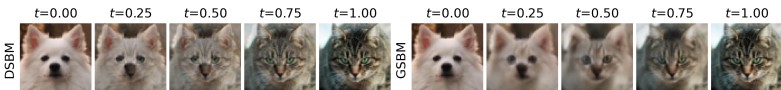

Figure 6: Mean of $p_t(X_t|x_0, x_1)$. Instead of using linear interpolation as in DSBM, GSBM *optimizes* $p_{t|0,1}$ w.r.t. (6) where $V_t$ is defined via a latent space, thereby exhibiting **more semantically meaningful interpolations**.

Table 2: FID values of dog→cat for DSBM and our GSBM.

| DSBM | GSBM |
|-------|-------|
| 14.16 | **12.39** |

## 4.2 IMAGE DOMAIN INTERPOLATION AND UNPAIRED TRANSLATION

Next, we consider unpaired translation between dogs and cats from AFHQ (Choi et al., 2020). We aim to explore how appropriate choices of state cost $V_t$ can help encourage more natural interpolations and more semantically meaningful couplings. While the design of $V_t$ itself is an interesting open question, here we exploit the geometry of a learned latent space. To this end, we use a pretrained variational autoencoder (Kingma & Welling, 2014), then define $V_t$ conditioned on an interpolation of two end points (notice that $x$ and $z$ respectively belong to image and latent spaces):

$$V_t(x|x_0, x_1) = \|x - \text{Decoder}(z_t)\|^2, \quad z_t := I(t, \text{Encoder}(x_0), \text{Encoder}(x_1)). \quad (14)$$

Though $I(t, z_0, z_1)$ can be any appropriate interpolation, we find the spherical linear interpolation (Shoemake, 1985) to be particularly effective, due to the Gaussian geometry in latent space. As the high dimensionality greatly impedes DeepGSB, we mainly compare with DSBM (Shi et al., 2023), the special case of our GSBM when $V_t$ is degenerate. As in DSBM, we resize images to 64×64.

Figure 5 reports the qualitative comparison between the generation processes of DSBM and our GSBM, along with their coupling during training. It is clear that, with the aid of a semantically meaningful $V_t$, GSBM typically converges to near-optimal coupling early in training, and, as expected, yields more interpretable generation processes. Interestingly, despite being subject to the same noise level ($\sigma$=0.5 in this case), the GSBM's generation processes are generally less noisy than DSBM. This is due to the optimization of the CondSOC problem (6) with our specific choice of $V_t$. As shown in Fig. 6, the conditional density $p_t(X_t|x_0, x_1)$ used in GSBM appears to eliminate unnatural artifacts observed in Brownian bridges, which rely on simple linear interpolation in pixel space. Quantitatively, our GSBM also achieves lower FID value as a measure of feasibility, as shown in Table 2. Finally, we note that the inclusion of $V_t$ and the solving of CondSOC increase wallclock time by *a mere 0.5%* compared to DSBM (see Table 5).

## 4.3 HIGH-DIMENSIONAL OPINION DEPOLARIZATION

Finally, we consider high-dimensional opinion depolarization, initially introduced in DeepGSB, where an opinion $X_t \in \mathbb{R}^{1000}$ is influenced by a polarizing effect (Schweighofer et al., 2020) when evolving through interactions with the population (see Appendix D.2):

$$dX_t = f_{\text{polarize}}(X_t; p_t)dt + \sigma dW_t \quad (15)$$

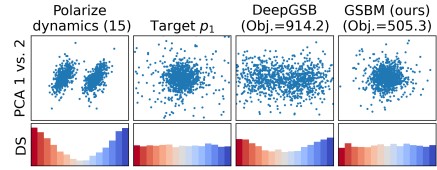

Figure 7: Distribution of terminal opinion $X_1 \in \mathbb{R}^{1000}$ and their directional similarities (DS; see Appendix D.2).

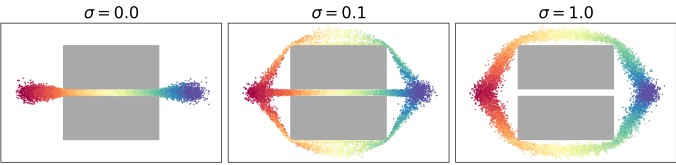

Figure 8: Our GSBM reveals how the optimal solution drastically changes w.r.t. the levels of noise ($\sigma$) when navigating through a narrow passway surrounded by the obstacles (shown in gray).

Table 3: Relative runtime between combination of matching losses and PI resampling on solving Stunnel, relative to $\mathcal{L}_{\text{explicit}}$.

|  | $\mathcal{L}_{\text{explicit}}$ | $\mathcal{L}_{\text{implicit}}$ |
|---|---|---|
| without PI | 100% | 276% |
| with PI | 108% | 284% |

Table 4: How the objective value (3) changes when enabling PI resampling on each matching loss and $\sigma$ in Stunnel. Performance is improved by deceasing ($-$) the objective values.

|  | $\sigma = 0.5$ | 1.0 | 2.0 |
|---|---|---|---|
| $\mathcal{L}_{\text{explicit}}$ | $-44.6 \pm 5.9$ | $-5.2 \pm 2.6$ | $1.5 \pm 3.5$ |
| $\mathcal{L}_{\text{implicit}}$ | $-112.4 \pm 53.7$ | $-2.0 \pm 8.0$ | $-0.7 \pm 6.0$ |

Table 5: Percentage of time spent in different stages of Alg. 5, measured on the AFHQ task.

| `match` (line 3) | Simulate $u_t^\theta$ (line 4) | Solve (6) (lines 5-6) |
|---|---|---|
| 64.3% | 35.2% | 0.5% |

Without any intervention, the opinion dynamics in (15) tend to segregate into groups with diametrically opposed views (first column of Fig. 7), as opposed to the desired unimodal distribution $p_1$ (second column of Fig. 7). To adapt our GSBM for this task, we treat (15) as a base drift, specifically defining $u_t^\theta(X_t) := f_{\text{polarize}}(X_t; p_t) + v_t^\theta(X_t)$, and then solving the CondSOC (6) by replacing the kinetic energy with $\int \|v_t^\theta\|^2 dt$. Similar to DeepGSB, we consider the same congestion cost $V_t$ defined in (12). As shown in Fig. 7, both DeepGSB and our GSBM demonstrate the capability to mitigate opinion segregation. However, our GSBM achieves closer proximity to the target $p_1$, indicating stronger feasibility, and achieve almost half the objective value (3) relative to DeepGSB.

## 4.4 DISCUSSIONS

**Effect of noise ($\sigma$).** In the stochastic control setting (Theodorou et al., 2010), the task-specific value of $\sigma$ plays a crucial role in representing the uncertainty from environment or the error in executing control. The optimal control thus changes drastically depending on $\sigma$. Figure 8 demonstrates how our GSBM correctly resolves this phenomenon, on an example of the famous "drunken spider" problem discussed in Kappen (2005). In the absence of noise ($\sigma=0$), it is very easy to steer through the narrow passage. When large amounts of noise is present ($\sigma=1.0$), there is a high chance of colliding with the obstacles, so the optimal solution is to completely steer around the obstacles.

**Ablation study on path integral (PI) resampling.** In Table 4, we ablate how the objective value changes when enabling PI resampling on different matching losses and noise $\sigma$ and report their performance (averaged over 5 independent trails) on the Stunnel task. We observe that PI resampling tends to enhance overall performance, particularly in low noise conditions, at the expense of a slightly increased runtime of 8%. Meanwhile, as shown in Table 3, implicit matching (4) typically requires longer time overall ($\times 2.7$ even in two dimensions), compared to its explicit counterpart.

**Profiling GSBM.** The primary algorithmic distinction between our GSBM and previous SB matching methods (Shi et al., 2023; Peluchetti, 2023) lies in how $p_{t|0,1}$ and $u_{t|0,1}$ are computed (see Lemma 3), where GSBM involves solving an additional CondSOC problem, *i.e.,* lines 5-6 in Alg. 5. Table 5 suggests that these computations, uniquely attached to GSBM, induce little computational overhead compared to other components in Alg. 5. This computational efficiency is due to our efficient variational approximation admitting simulation-free, parallelizable optimization.

## 5 CONCLUSION AND LIMITATION

We developed GSBM, a new matching algorithm that provides approximate solutions to the Generalized Schrödinger Bridge (GSB) problems. We demonstrated strong capabilities of GSBM over prior methods in solving crowd navigation, opinion modeling, and interpretable domain transfer. It should be noticed that GSBM requires differentiability of $V_t$ and relies on the quadratic control cost to establish its convergence analysis, which, despite notably improving over prior methods, remains as necessary conditions. We acknowledge these limitations and leave them for future works.

ACKNOWLEDGEMENTS

We acknowledge the Python community (Van Rossum & Drake Jr, 1995; Oliphant, 2007) and the core set of tools that enabled this work, including PyTorch (Paszke et al., 2019), functorch (Horace He, 2021), torchdiffeq (Chen, 2018), JAX (Bradbury et al., 2018), Flax (Heek et al., 2020), Hydra (Yadan, 2019), Jupyter (Kluyver et al., 2016), Matplotlib (Hunter, 2007), numpy (Oliphant, 2006; Van Der Walt et al., 2011), and SciPy (Jones et al., 2014).

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

## A    REVIEWS ON STOCHASTIC OPTIMAL CONTROL AND RELATED WORKS

**Solving Generalized Schrödinger bridge by reframing as stochastic optimal control**    The solution to a Generalized Schrödinger Bridge (GSB) problem (3) can also be expressed as the solution a stochastic optimal control (SOC) problem, typically structured as

$$\min_{u_t(\cdot)} \int_0^1 \mathbb{E}_{p_t} \left[ \frac{1}{2} \|u_t(X_t)\|^2 + V_t(X_t) \right] \mathrm{d}t + \mathbb{E}_{p_1} \left[ \phi(X_1) \right], \tag{16a}$$

$$\text{s.t. } \mathrm{d}X_t = u_t(X_t)\mathrm{d}t + \sigma \mathrm{d}W_t, \quad X_0 \sim \mu, \tag{16b}$$

where we see that the terminal distribution "hard constraint" in GSB (3) is instead relaxed into a soft "terminal cost" $\phi(\cdot) : \mathbb{R}^d \to \mathbb{R}$. Problems with the forms of either (16) or (3) are known to be tied to linearly-solvable Markov decision processes (Todorov, 2007; Rawlik et al., 2013), corresponding to a *tractable* class of SOC problems whose optimality conditions—the Hamilton–Jacobi–Bellman equations—admit efficient approximation. This is attributed to the presence of the $\ell_2$-norm control cost, which can be interpreted as the KL divergence between controlled and uncontrolled processes. The interpretation bridges the SOC problems to probabilistic inference (Levine, 2018; Okada & Taniguchi, 2020), from which machine learning algorithms, such as our GSBM, can be developed.

However, naïvely transforming GSB problems (3) into SOC problems can introduce many potential issues. The design of the terminal cost is extremely important, and in many cases, it is an *intractable cost function* as we do not have access to the densities $\mu$ and $\nu$. Prior works have mainly stuck to simple terminal costs (Ruthotto et al., 2020), using biased approximations based on batch estimates (Koshizuka & Sato, 2023), or using an adversarial approach to learn the cost function (Zhu et al., 2017; Lin et al., 2021). Furthermore, this approach will necessitate differentiating through an SDE simulation, requiring *high memory usage*. Though memory-efficient adjoint methods have been developed (Chen et al., 2018; Li et al., 2020), they remain *computationally expensive* to use at scale.

In contrast, our GBSM approach only requires samples from $\mu$ and $\nu$. By enforcing boundary distributions as a hard constraint instead of a soft one, our algorithm finds solutions that satisfy feasibility much better in practice. This further allows us to consider higher dimensional problems without the need to introduce additional hyperparameters for tuning a terminal cost. Finally, our algorithm drastically reduces the number of SDE simulations required, as both the matching algorithm (Stage 1) and the CondSOC variational formulation (Stage 2) of GSBM can be done simulation-free.

**Comparison to related works**    Table 6 compares to prior learning based methods that also provide approximate solutions to (3) to our GSBM. Meanwhile, Fig. 9 summarizes different classes of matching algorithms.

Table 6: Our **GSBM** features better feasibility, requires only samples from $\mu, \nu$, has local convergence analysis, and exhibits much better scalability.

| | Feasibility (2) | Requirement from distributions $\mu, \nu$ | Convergence analysis | Dimension $d$ |
|---|---|---|---|---|
| NLSB (Koshizuka & Sato, 2023) | ✗(relaxed) | samples & densities | ✓(local) | 5 |
| DeepGSB (Liu et al., 2022) | ≈ **F** only in limit | samples & densities | ✗ | 1000 |
| **GSBM** (this work) | ≈ **F** | only samples | ✓(local) | ≳ 12K |

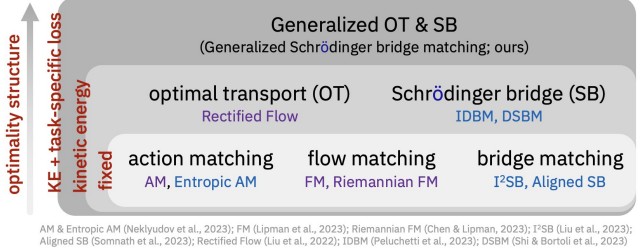

AM & Entropic AM (Neklyudov et al., 2023); FM (Lipman et al., 2023); Riemannian FM (Chen & Lipman, 2023); I²SB (Liu et al., 2023); Aligned SB (Somnath et al., 2023); Rectified Flow (Liu et al., 2022); IDBM (Peluchetti et al., 2023); DSBM (Shi & Bortoli et al., 2023)

Figure 9: Summary of different matching algorithms. Notice that OT and SB are w.r.t. $\ell_2$ costs.

## B  Proofs

**Proposition 1** (Stage 1). *The unique minimizer to Stage 1 coincides with* $\nabla s_t^\star(X_t)$.

*Proof.* By keeping $p_t$ fixed in (3), the value of $\mathbb{E}_{p_t}[V_t(X_t)]$ no longer relies on $u_t^\theta$, rendering (3) the same optimization problem as in the implicit matching; see (4) and Sec. 2. As the implicit matching (4) admits a unique minimizer $\nabla s_t^\star$, we conclude the proof. □

**Proposition 2** (Stage 2; Conditional stochastic optimal control; CondSOC). *Let the minimizer to Stage 2 be factorized by $p_t(X_t) = \int p_t(X_t|x_0, x_1)p_{0,1}^\theta(x_0, x_1)\mathrm{d}x_0\mathrm{d}x_1$ where $p_{0,1}^\theta$ is the boundary coupling induced by solving (1) with $u_t^\theta$. Then, $p_t(X_t|x_0, x_1) \equiv p_{t|0,1}$ solves :*

$$\min_{u_{t|0,1}} \mathcal{J} := \int_0^1 \mathbb{E}_{p_t(X_t|x_0,x_1)} \left[ \frac{1}{2}\|u_t(X_t|x_0,x_1)\|^2 + V_t(X_t) \right] \mathrm{d}t \tag{6a}$$

$$\text{s.t. } \mathrm{d}X_t = u_t(X_t|x_0,x_1)\mathrm{d}t + \sigma\mathrm{d}W_t, \quad X_0 = x_0, \; X_1 = x_1 \tag{6b}$$

*Proof.* Let us recall the GSB problem (3) in the form of FPE constraint (2):

$$\min_{u_t} \int_0^1 \mathbb{E}_{p_t} \left[ \frac{1}{2}\|u_t(X_t)\|^2 + V_t(X_t) \right] \mathrm{d}t \tag{17a}$$

$$\text{s.t. } \frac{\partial}{\partial t}p_t(X_t) = -\nabla \cdot (u_t(X_t)\, p_t(X_t)) + \frac{\sigma^2}{2}\Delta p_t(X_t), \quad p_0 = \mu, p_1 = \nu. \tag{17b}$$

Under mild regularity assumptions (Anderson, 1982; Yong & Zhou, 1999) such that Leibniz rule and Fubini's Theorem apply, we can separate $p_{0,1}$—as it is assumed to be fixed—out of the marginal $p_t$. Specifically, the objective value (17a) can be decomposed into

$$\int_0^1 \mathbb{E}_{p_t} \left[ \frac{1}{2}\|u_t\|^2 + V_t \right] \mathrm{d}t = \int_0^1 \mathbb{E}_{p_{0,1}} \left[ \mathbb{E}_{p_{t|0,1}} \left[ \frac{1}{2}\|u_t\|^2 + V_t \right] \right] \mathrm{d}t, \quad \text{by law of total expectation}$$

$$= \mathbb{E}_{p_{0,1}} \left[ \int_0^1 \mathbb{E}_{p_{t|0,1}} \left( \frac{1}{2}\|u_t\|^2 + V_t \right) \mathrm{d}t \right], \quad \text{by Fubini's Theorem}$$

which recovers (6a). Similarly, each term in the FPE (17b) can be decomposed into

$$\frac{\partial}{\partial t}p_t(X_t) = \frac{\partial}{\partial t}\int p_t(X_t|x_0,x_1)p_{0,1}(x_0,x_1)\mathrm{d}x_{0,1} = \mathbb{E}_{p_{0,1}} \left[ \frac{\partial}{\partial t}p_{t|0,1} \right],$$

$$\nabla \cdot (u_t(X_t)\, p_t(X_t)) = \nabla \cdot \left( u_t(X_t)\int p_t(X_t|x_0,x_1)p_{0,1}(x_0,x_1)\mathrm{d}x_{0,1} \right)$$

$$= \nabla \cdot \left( \int u_t(X_t)p_t(X_t|x_0,x_1)p_{0,1}(x_0,x_1)\mathrm{d}x_{0,1} \right)$$

$$= \mathbb{E}_{p_{0,1}} \left[ \nabla \cdot \left( u_t\, p_{t|0,1} \right) \right],$$

$$\Delta p_t(X_t) = \nabla \cdot (\nabla p_t(X_t))$$

$$= \nabla \cdot \left( \nabla \int p_t(X_t|x_0,x_1)p_{0,1}(x_0,x_1)\mathrm{d}x_{0,1} \right)$$

$$= \mathbb{E}_{p_{0,1}} \left[ \nabla \cdot \nabla p_{t|0,1} \right]$$

$$= \mathbb{E}_{p_{0,1}} \left[ \Delta p_{t|0,1} \right],$$

and, finally, $p_0(x_0) = \mathbb{E}_{p_{0,1}}[\delta_{x_0}(x)]$ and $p_1(x_1) = \mathbb{E}_{p_{0,1}}[\delta_{x_1}(x)]$. Collecting all related terms yields:

$$\frac{\partial}{\partial t}p_{t|0,1} = -\nabla \cdot \left( u_t\, p_{t|0,1} \right) + \Delta p_{t|0,1}, \quad p_0 = \delta_{x_0}, p_1 = \delta_{x_1},$$

which is equivalent to the conditional SDE in (6b). Note that we denote $u_t$ as $u_t(X_t|x_0,x_1)$ to emphasize the fact that when $p_{t|0,1}(X_t|x_0,x_1)$ is factorized out of $p_t(X_t)$, the GSB problem (3) can be factorized into a mixture of SOC problems, each with an end-point constraint $(x_0, x_1)$. □

**Remark** (PDE interpretation of Proposition 2 and (6)). The optimal control to (6) is given by $u_t^\star(X_t|x_0, x_1) = \sigma^2 \nabla \log \Psi_t(X_t|x_1)$, where the time-varying potential $\Psi_t(X_t|x_1)$ solves a partial differential equation (PDE) known as the Hamilton-Jacobi-Bellman (HJB) PDE:

$$\frac{\partial}{\partial t}\Psi_t(x|x_1) = -\frac{1}{2}\sigma^2 \Delta \Psi_t(x|x_1) + V_t(x)\Psi_t(x|x_1), \quad \Psi_1(x|x_1) = \delta_{x_1}(x). \tag{18}$$

In general, (18) lacks closed-form solutions, except in specific instances. For example, when $V$ is quadratic, the solution is provided in (19). Additionally, when $V$ is degenerate, (18) simplifies to a heat kernel, and its solution corresponds to the drift of the Brownian bridge utilized in DSBM (Shi et al., 2023). Otherwise, one can approximate its solution with the aid of path-integral theory, as shown in Prop. 4).

**Lemma 3** (Analytic solution to (6) for quadratic $V$ and $\sigma > 0$). *Let $V(x) := \alpha\|\sigma x\|^2$, $\alpha, \sigma > 0$, then the optimal solution to (6) follows a Gaussian path $X_t^\star \sim \mathcal{N}(c_t x_0 + e_t x_1, \gamma_t^2 \mathbf{I}_d)$, where*

$$c_t = \frac{\sinh(\eta(1-t))}{\sinh \eta}, \quad e_t = \cosh(\eta(1-t)) - c_t \cosh \eta, \quad \gamma_t = \sigma\sqrt{\frac{\sinh(\eta(1-t))}{\eta}e_t}, \quad \eta = \sigma\sqrt{2\alpha}.$$

*These coefficients recover Brownian bridges as $\alpha \to 0$ and, if further $\sigma \to 0$, straight lines.*

*Proof.* This is a direct consequence of conditioned diffusion processes with quadratic killing rates (Mazzolo & Monthus, 2022). Specifically, Mazzolo & Monthus (2022, Eq. (84)) give the analytic expression to the optimal control of (6), when $V_t := \alpha\|\sigma x\|^2$:

$$u_t(X_t|x_0, x_1) = \frac{\eta}{\sinh(\eta(1-t))}x_1 - \frac{\eta}{\tanh(\eta(1-t))}X_t, \quad \eta := \sigma\sqrt{2\alpha}. \tag{19}$$

As (19) suggests a linear SDE, its mean $\mu_t \in \mathbb{R}^d$ solves an ODE (Särkkä & Solin, 2019):

$$\frac{d\mu_t}{dt} = \frac{\eta}{\sinh(\eta(1-t))}x_1 - \frac{\eta}{\tanh(\eta(1-t))}\mu_t$$

whose analytic solution is given by

$$\mu_t = \frac{x_1 \int_0^t \left(\frac{\eta g_\tau}{\sinh(\eta(1-\tau))}\right)d\tau + K}{g_t}, \tag{20}$$

where $K$ depends on the initial condition and

$$g_t := \exp\left(\int_0^t \frac{\eta}{\tanh(\eta(1-\tau))}d\tau\right) \stackrel{(*)}{=} \exp\left([\ln v_\tau]_{\sinh(\eta(1-t))}^{\sinh \eta}\right) = \frac{\sinh \eta}{\sinh(\eta(1-t))}.$$

Note that $(*)$ is due to change of variable $v_\tau := \sinh(\eta(1-\tau))$. Substituting $g_t$ back to (20), and noticing that $x_0 = \mu_0 = \frac{x_1 \cdot 0 + K}{1} = K$, yields the desired coefficients:

$$\mu_t = c_t x_0 + e_t x_1, \quad c_t = \frac{\sinh(\eta(1-t))}{\sinh \eta}, \quad e_t = \cosh(\eta(1-t)) - c_t \cosh \eta. \tag{21}$$

Similarly, the variance $\Sigma_t \in \mathbb{R}^{d\times d}$ of the SDE in (19) solves an ODE

$$\frac{d\Sigma_t}{dt} = -\frac{2\eta}{\tanh(\eta(1-t))}\Sigma_t + \sigma^2 \mathbf{I}_d.$$

Repeating similar derivation, as in solving $\mu_t$, leads to

$$\Sigma_t = \frac{\sigma^2}{\eta}\sinh^2(\eta(1-t))\left(\coth(\eta(1-t)) - \coth \eta\right)\mathbf{I}_d = \frac{\sigma^2}{\eta}\sinh(\eta(1-t))e_t\mathbf{I}_d. \tag{22}$$

which gives the desired $\gamma_t$ since $\Sigma_t = \gamma_t^2 \mathbf{I}_d$.

We now show how these coefficients (21) and (22) recover Brownian bridge as $\alpha$ or, equivalently, $\eta := \sigma\sqrt{2\alpha}$ approaches the zero limit.

$$\lim_{\eta \to 0} c_t = \lim_{\eta \to 0} \frac{e^{\eta(1-t)} - e^{-\eta(1-t)}}{e^\eta - e^{-\eta}} \stackrel{(*)}{=} \frac{1 + \eta(1-t) - (1 - \eta(1-t))}{1 + \eta - (1 - \eta)} = 1 - t,$$

$$\lim_{\eta \to 0} e_t = \lim_{\eta \to 0}\left(\cosh(\eta(1-t)) - c_t \cosh \eta\right) = 1 - (1-t)\cdot 1 = t,$$

$$\lim_{\eta \to 0} \gamma_t = \lim_{\eta \to 0}\sigma\sqrt{\frac{\sinh(\eta(1-t))}{\eta}e_t} \stackrel{(**)}{=} \sigma\sqrt{(1-t)t},$$

where $(*)$ and $(**)$ are respectively due to $\exp(x) \approx 1 + x$ and

$$\lim_{\eta \to 0} \frac{\sinh(\eta(1-t))}{\eta} = \lim_{\eta \to 0} \frac{e^{\eta(1-t)} - e^{-\eta(1-t)}}{2\eta} \overset{(*)}{=} \frac{1 + \eta(1-t) - (1 - \eta(1-t))}{2\eta} = 1 - t.$$

Hence, we recover the analytic solution to Brownian bridge. It can be readily seen that, if further $\sigma \to 0$, the solution collapses to linear interpolation between $x_0$ and $x_1$. □

**Remark** (Lemma 3 satisfies the boundary conditions in (6b)). One can verify that $c_0 = 1$, $e_0 = 0$, and $\gamma_0 = 0$. Furthermore, we have $c_1 = 0$, $e_1 = 1$, and $\gamma_1 = 0$. Hence, as expected, Lemma 3 satisfies the boundary condition in (6b).

**Proposition 4** (Path integral solution to (6)). *Let $r(\bar{X}|x_0, x_1)$ be a distribution absolutely continuous w.r.t. the Brownian motion, denoted $q(\bar{X}|x_0)$, where we shorthand $\bar{X} \equiv X_{t \in [0,1]}$. Suppose $r$ is associated with an SDE $dX_t = v_t(X_t)dt + \sigma dW_t$, $X_0 = x_0$, $X_1 = x_1$, $\sigma > 0$. Then, the optimal, path-integral solution to (6) can be obtained via*

$$p^\star(\bar{X}|x_0, x_1) = \tfrac{1}{Z}\omega(\bar{X}|x_0, x_1)r(\bar{X}|x_0, x_1), \tag{10}$$

*where $Z$ is the normalization constant and $\omega$ is the importance weight:*

$$\omega(\bar{X}|x_0, x_1) := \exp\left(-\int_0^1 \frac{1}{\sigma^2}\left(V_t(X_t) + \frac{1}{2}\|v_t(X_t)\|^2\right)dt - \int_0^1 \frac{1}{\sigma}v_t(X_t)^\top dW_t\right). \tag{11}$$

*Proof.* As the terminal boundary condition of (6b) is pinned at $x_1$, we can transform (6) into:

$$\min_{u_{t|0,1}} \int_0^1 \mathbb{E}_{p_t(X_t|x_0, x_1)}\left[\frac{1}{2\sigma^2}\|u_t(X_t|x_0, x_1)\|^2 + \frac{1}{\sigma^2}V_t(X_t)\right]dt - \log \mathbb{1}_{x_1}(X_1), \tag{23a}$$

$$\text{s.t. } dX_t = u_t(X_t|x_0, x_1)dt + \sigma dW_t, \quad X_0 = x_0 \tag{23b}$$

where $\mathbb{1}_{\mathcal{A}}(\cdot)$ is the indicator function of the set $\mathcal{A}$. Hence, the terminal cost "$-\log \mathbb{1}_{x_1}(x)$" vanishes at $x = x_1$ but otherwise explodes. Equation (23) is a valid stochastic optimal control (SOC) problem, and its optimal solution can be obtained as the form of path integral (Kappen, 2005):

$$p^\star(\bar{X}|x_0, x_1) = \frac{1}{Z'}\exp\left(-\int_0^1 \frac{1}{\sigma^2}V_t(X_t)\,dt\right)\mathbb{1}_{x_1}(X_1)q(\bar{X}|x_0), \tag{24}$$

where $q(\cdot|x_0)$ is the density of the Brownian motion conditioned on $X_0 = x_0$, the normalization constant $Z'$ is such that (24) remains as a proper distribution, and we shorthand $\bar{X} \equiv X_{t \in [0,1]}$. We highlight that Equations (23) and (24) are essential transformation that allow us to recover the conditional distribution used in prior works (Shi et al., 2023) when $V_t := 0$ (see the remark below)

Directly re-weighting samples from $q$ using (24) may have poor complexity as most samples are assigned with zero weights due to $\mathbb{1}_{x_1}(\cdot)$. A more efficient alternative is to rebase the sampling distribution in (24) from $q$ to an importance sampling $r(\bar{X}|x_0, x_1)$ such that $X_0 = x_0$, $X_1 = x_1$, and is absolutely continuous with respect to $q$. Then, the remarkable results from information-theoretic SOC (Theodorou et al., 2010; Theodorou, 2015) suggest

$$p^\star(\bar{X}|x_0, x_1) = \frac{1}{Z}\exp\left(-\int_0^1 \frac{1}{\sigma^2}V_t(X_t)\,dt\right)\frac{q(\bar{X}|x_0)}{r(\bar{X}|x_0, x_1)}r(\bar{X}|x_0, x_1), \tag{25}$$

where the Radon-Nikodym derivative $\frac{q}{r}$ can be computed via Girsanov's theorem (Särkkä & Solin, 2019):

$$\frac{q(\bar{X}|x_0)}{r(\bar{X}|x_0, x_1)} = \exp\left(-\int_0^1 \frac{1}{2\sigma^2}\|v_t(X_t)\|^2 dt - \int_0^1 \frac{1}{\sigma}v_t(X_t)^\top dW_t\right). \tag{26}$$

Substituting (26) back to (25) concludes the proof. □

**Remark** (Equation (24) recovers Brownian bridge when $V_t := 0$). One can verify that, when $V_t := 0$, the optimal solution $p^\star(\bar{X}|x_0, x_1) \propto q(\bar{X}|x_0)\mathbb{1}_{x_1}(X_1) = q(\bar{X}|x_0, x_1)$ is simply the Brownian motion *conditioned* on the end point being $x_1$, which is precisely the Brownian bridge.

**Theorem 5** (Local convergence). *Let the intermediate result of Stage 1 and 2, after repeating $n$ times, be $\theta^n$. Then, its objective value in ($3$), $\mathcal{L}(\theta^n)$, is monotonically non-increasing as $n$ grows:*

$$\mathcal{L}(\theta^n) \geq \mathcal{L}(\theta^{n+1}).$$

*Proof.* Let $p_t^{\theta^n}$ and $u_t^{\theta^n}$ be the marginal distribution and vector field, respectively, after $n$ steps of alternating optimization according to Alg. $5$, *i.e.*, $p_t^{\theta^n}$ and $u_t^{\theta^n}$ satisfy the FPE. Furthermore, let $p_{0,1}^{\theta^n}$ and $p_{t|0,1}^{\theta^n}$ be the coupling and conditional distribution, respectively, defined by $u_t^{\theta^n}$, and let $p_{t|0,1}^{\star}$ be the solution to CondSOC ($6$). This implies that the marginal distributions at step $n+1$ are given by

$$p_t^{\theta^{n+1}} := \int p_{0,1}^{\theta^n} p_{t|0,1}^{\star} \mathrm{d}x_{0,1}. \tag{27}$$

Now, under mild assumptions (Anderson, 1982; Yong & Zhou, 1999) such that all distributions approach zero at a sufficient speed as $\|x\| \to \infty$, and that all integrands are bounded, we have (note that inputs to all functions are dropped for notational simplicity):

$$
\begin{aligned}
\mathcal{L}(\theta^n) &= \int\int p_t^{\theta^n} \left( \frac{1}{2}\|u_t^{\theta^n}\|^2 + V_t \right) \mathrm{d}x_t \mathrm{d}t \\
&= \int p_{0,1}^{\theta^n} \left[ \int\int p_{t|0,1}^{\theta^n} \left( \frac{1}{2}\|u_t^{\theta^n}\|^2 + V_t \right) \mathrm{d}x_{t|0,1}\mathrm{d}t \right] \mathrm{d}x_{0,1} \quad \text{by Fubini's Theorem} \\
&\geq \int p_{0,1}^{\theta^n} \left[ \int\int p_{t|0,1}^{\star} \left( \frac{1}{2}\|u_t^{\theta^n}\|^2 + V_t \right) \mathrm{d}x_{t|0,1}\mathrm{d}t \right] \mathrm{d}x_{0,1} \quad \text{by optimizing ($6$)} \\
&= \int\int p_t^{\theta^{n+1}} \left( \frac{1}{2}\|u_t^{\theta^n}\|^2 + V_t \right) \mathrm{d}x_t \mathrm{d}t \quad\quad\quad\quad\quad \text{by Fubini's Theorem and ($27$)} \\
&\geq \int\int p_t^{\theta^{n+1}} \left( \frac{1}{2}\|u_t^{\theta^{n+1}}\|^2 + V_t \right) \mathrm{d}x_t \mathrm{d}t \tag{28} \\
&= \mathcal{L}(\theta^{n+1}),
\end{aligned}
$$

where we factorize the solution of Stage 1 by $p_t^{\theta^n} := \int p_{0,1}^{\theta^n} p_{t|0,1}^{\theta^n} \mathrm{d}x_{0,1}$. The inequality in ($28$) follows from the fact that $u_t^{\theta^{n+1}}$, as the solution to the proceeding Stage 1, finds the unique minimizer that yields the same marginal as $p_t^{\theta^{n+1}}$ while minimizing kinetic energy, *i.e.*,

$$\int p_t^{\theta^{n+1}} \|u_t^{\theta^n}\|^2 \,\mathrm{d}x_t \geq \int p_t^{\theta^{n+1}} \|u_t^{\theta^{n+1}}\|^2 \,\mathrm{d}x_t.$$

$\square$

**Theorem 6.** *The optimal solution to GSB problem ($3$) is a fixed point of GSBM, Alg. $5$.*

*Proof.* Let $p_t^{\star}$ and $u_t^{\star}$ be the optimal solution to the GSB problem in ($3$), and suppose $p_t^{\star} := \int p_{t|0,1}^{\star} p_{0,1}^{\star} \mathrm{d}x_0 \mathrm{d}x_1$. It suffices to show that

1. Given $x_0, x_1 \sim p_{0,1}^{\star}$, the conditional distribution $p_{t|0,1}^{\star}$ is the optimal solution to ($6$).

2. Given $p_t^{\star}$, both matching algorithms (Alg. $1$ and $2$) return $u_t^{\star}$.

The first statement follows directly from the Forward-Backward SDE (FBSDE) representation of ($3$), initially derived in Liu et al. (2022, Theorem 2). The FBSDE theory suggests that the (conditional) optimal control bridging any $x_0, x_1 \sim p_{0,1}^{\star}$ satisfies a BSDE (Pardoux & Peng, 2005) that is uniquely associated with the HJB PDE of ($6$), *i.e.*, ($18$). This readily implies that $p_{t|0,1}^{\star}$ is the solution to ($6$).

Since $u_t^{\star}$ is known to be a gradient field (Liu et al., 2022, Eq. (3)), it must be with the solution returned by the implicit matching algorithm (Alg. $1$), as $\mathcal{L}_{\text{implicit}}$ returns the unique gradient field that matches $p_t^{\star}$. On the other hand, since the explicit matching loss ($5$) can be interpreted as a Markovian projection (Shi et al., 2023), *i.e.*, it returns the closest (in the KL sense) Markovian process to the reciprocal path measure (Léonard, 2013; Léonard et al., 2014) defined by the solution to ($6$). Since

the solution to (6), as proven in the first statement, is simply $p_{t|0,1}^{\star}$, the closest Markovian process is by construction $u_t^{\star}$. Hence, the second statement also holds, and we conclude the proof. It is important to note that, while $\mathcal{L}_{\text{explicit}}$ and $\mathcal{L}_{\text{implicit}}$ do not share the same minimizer in general, they do at the equilibrium $p_t^{\star}$. □

## C  ADDITIONAL DERIVATIONS AND DISCUSSIONS

### C.1  EXPLICIT MATCHING LOSS

**How (5) preserves the prescribed $p_t$.**  A rigorous derivation of (5) can be found in, *e.g.,* Shi et al. (2023, Proposition 2), called *Markovian projection*. Here, we provide an alternative derivation that follows closer to the one from flow matching (Lipman et al., 2023).

**Lemma 7.** *Let the marginal $p_t$ be constructed from a mixture of conditional probability paths, i.e., $p_t(x) := \mathbb{E}_{p_{0,1}}[p_t(x|x_0, x_1)]$, where $p_t(x|x_0, x_1) \equiv p_{t|0,1}$ is the time marginal of the SDE, $\mathrm{d}X_t = u_t(X_t|x_0, x_1)\mathrm{d}t + \sigma\mathrm{d}W_t,\ X_0 = x_0,\ X_1 = x_1$, then the SDE drift that satisfies the FPE prescribed by $p_t$ is given by*

$$u_t^{\star}(x) = \frac{\mathbb{E}_{p_{0,1}}[u_t(x|x_0, x_1)p_{t|0,1}(x|x_0, x_1)]}{p_t(x)}. \tag{29}$$

*Proof.* It suffices to check that $u_t^{\star}(x)$ satisfies the FPE prescribed by $p_t$:

$$\begin{aligned}
\frac{\partial}{\partial t}p_t(x) &= \int \left(\frac{\partial}{\partial t}p_{t|0,1}\right)p_{0,1}\mathrm{d}x_{0,1} \\
&= \int \left(-\nabla \cdot \left(u_{t|0,1}\ p_{t|0,1}\right)\right)p_{0,1}\mathrm{d}x_{0,1} + \int \left(\frac{1}{2}\sigma^2\Delta p_{t|0,1}\right)p_{0,1}\mathrm{d}x_{0,1} \\
&\overset{(29)}{=} -\nabla \cdot (u_t^{\star}\ p_t) + \frac{1}{2}\sigma^2\Delta p_t.
\end{aligned}$$

□

An immediate consequence of Lemma 7 is that

$$\mathcal{L}_{\text{explicit}}(\theta) = \mathbb{E}_{p_t}\left[\frac{1}{2}\|u_t^{\theta}(X_t) - u_t^{\star}(X_t)\|^2\right] + \mathcal{O}(1), \tag{30}$$

where $\mathcal{O}(1)$ is independent of $\theta$. Hence, the $u_t^{\theta^{\star}}$ preserves the prescribed $p_t$.

**Relation to implicit matching (4).**  Both explicit and implicit matching losses are associated to some regression objectives, except w.r.t. different targets, *i.e.,* $u_t^{\star}$ in (30) *vs.* $\nabla s_t^{\star}$ in Sec. 2. As pointed out in Neklyudov et al. (2023), the two targets relate to each other via the Helmholtz decomposition (Ambrosio et al., 2008, Lemma 8.4.2), which suggests that $u_t^{\star} = \nabla s_t^{\star} + w_t$ where $w_t$ is the divergence-free vector field, *i.e.,* $\nabla \cdot (w_t p_t) = 0$. Though this implies that $\mathcal{L}_{\text{explicit}}$ only upper-bounds the kinetic energy, its solution sequential, by alternating between solving (6) in Stage 2, remains well-defined from the measure perspective (Peluchetti, 2022; 2023; Shi et al., 2023). Specifically, the sequential performs alternate projection between reciprocal path measure defined by the solution to (6), in the form $p_t := \mathbb{E}_{p_{0,1}}[p_{t|0,1}]$, and the Markovian path measure, and admits convergence to the optimal solution for standard SB problems.

### C.2  GAUSSIAN PROBABILITY PATH

**Derivation of analytic conditional drift in (8).**  Recall the Gaussian path approximation in (7):

$$X_t = \mu_t + \gamma_t Z, \quad Z \sim \mathcal{N}(0, \boldsymbol{I}_d),$$

which immediately implies the velocity vector field and the score function (Särkkä & Solin, 2019; Albergo et al., 2023):

$$\partial_t X_t = \partial_t \mu_t + \frac{\partial_t \gamma_t}{\gamma_t}\left(X_t - \mu_t\right), \quad \nabla \log p_t(X_t) = -\frac{1}{\gamma_t^2}\left(X_t - \mu_t\right).$$

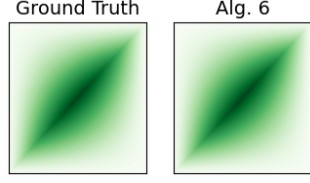

Figure 10: How Alg. 6 faithfully recovers the covariance matrix of, in this case, Brownian bridge.

**Algorithm 6** `CovSample` (line 1 in Alg. 4)

**Require:** SDE (6b) with $\mu_t, \gamma_t$
    Discretize $\bar{t} := [t_1, \cdots, t_K]$, $0 < t_1 < \cdots < t_K < 1$
    Solve $g_{\bar{t}}$ according the 1D ODE in (32)
    Compute covariance matrix $C_{\bar{t}} \in \mathbb{R}^{K \times K}$ by (33)
    Perform Cholesky decomposition $C_{\bar{t}} = L_{\bar{t}} L_{\bar{t}}^T$
    Sample $X_{\bar{t}} = \mu_{\bar{t}} + L_{\bar{t}} Z_{\bar{t}}$, $Z_{\bar{t}} \in \mathbb{R}^{K \times d}$, $[Z_{\bar{t}}]_{ij} \sim \mathcal{N}(0,1)$
    **return** $X_{\bar{t}}$

We can then construct the conditional drift:

$$u_t(X_t|x_0, x_1) = \partial_t X_t + \frac{\sigma^2}{2} \nabla \log p_t(X_t) = \partial_t \mu_t + \frac{1}{\gamma_t} \left( \partial_t \gamma_t - \frac{\sigma^2}{2\gamma_t} \right) (X_t - \mu_t). \tag{31}$$

Notice that (31) is of the form of a gradient field due to the linearity in $X_t$. One can verify that substituting the Brownian bridge, $\mu_t := (1-t)x_0 + tx_1$ and $\gamma_t := \sigma\sqrt{t(1-t)}$, to (31) indeed yields the desired drift $\frac{x_1 - X_t}{1-t}$.

**Efficient simulation with analytic covariance function (Footnote 4).** Proposition 4 requires samples from the "*joint*" distribution in path space. This requires sequential simulation, which can scale poorly to higher-dimensional applications. Fortunately, there exists efficient computation to (6b) with the (linear) conditional drift $u_{t|0,1}$ given by (8), as the analytic solution to (6b) reads

$$X_t = e^{g_t} \left( x_0 + \int_0^t e^{-g_\tau} b_\tau d\tau + \int_0^t e^{-g_\tau} \sigma dW_s \right), \quad g_t := \int_0^t a_\tau d\tau, \tag{32}$$

where $a_t := \frac{1}{\gamma_t} \left( \partial_t \gamma_t - \frac{\sigma^2}{2\gamma_t} \right) \in \mathbb{R}$ is defined as in (8), $b_t := \partial_t \mu_t - a_t \mu_t$, and $\int dW_s$ is the Ito stochastic integral (Itô, 1951). The covariance function between two time steps $s, t$, such that $0 \leq s < t \leq 1$, can then be computed by

$$
\begin{aligned}
\mathrm{Cov}(s,t) &= e^{g_s + g_t} \langle \int_0^s e^{-g_\tau} \sigma dW_\tau, \int_0^t e^{-g_\tau} \sigma dW_\tau \rangle \\
&\overset{(i)}{=} e^{g_s + g_t} \langle \int_0^s e^{-g_\tau} \sigma dW_\tau, \int_0^s e^{-g_\tau} \sigma dW_\tau \rangle \\
&\overset{(ii)}{=} e^{g_s + g_t} \int_0^s e^{-2g_\tau} \sigma^2 dt \\
&\overset{(iii)}{=} e^{g_t - g_s} \gamma_s^2,
\end{aligned}
$$

where $(i)$ is due to the independence of the Ito integral between $[0, s]$ and $[s, t]$, $(ii)$ is due to the Ito isometry, and, finally, $(iii)$ is due to substituting $\gamma_s^2 = \mathrm{Var}(s) := e^{2g_s} \int_0^s e^{-2g_\tau} \sigma^2 dt$.

Repeating the same derivation for $t, s$ such that $0 \leq t < s \leq 1$, the covariance function of (6b) between any given two timesteps $s, t \in [0, 1]$ can be written cleanly as

$$\mathrm{Cov}(s,t) = \gamma_{\min(s,t)}^2 e^{g_{\max(s,t)} - g_{\min(s,t)}}, \tag{33}$$

which, crucially, requires only solving an "1D" ODE, $dg_t = a_t dt, g_0 = 0$. We summarize the algorithm in Alg. 6 with an example of Brownian bridge in Fig. 10. We note again that all computation is parallelizable over the batches.

## C.3 ANALYTIC SOLUTION TO (6) FOR QUADRATIC $V$ AND $\sigma = 0$

Recall the finite-horizon and continuous-time linear quadratic regulator, defined as:

$$\min_{u_t} x_1^\top F x_1 + \int_0^1 \left[ x_t^\top Q x_t + u_t^\top R u_t \right] dt \quad \text{s.t. } \dot{x}_t = Ax_t + Bu_t, \tag{34}$$

whose optimal control $u_t^\star = -R^{-1}BP_t x_t$ can be obtained by solving a Riccati differential equation:

$$-\dot{P}_t = A^\top P_t + P_t A - P_t B R^{-1} B^\top P_t + Q, \quad P_1 = F,$$

or, equivalently, a Lyapunov differential equation (one can verify that $H_t = P_t^{-1}$ for all $t$):

$$\dot{H}_t = AH_t + H_t A^\top - BR^{-1}B^\top + H_t Q H_t, \quad H_1 = F^{-1}. \tag{35}$$

The end-point constraint can be accounted by setting $F \to \infty$, as suggested in Chen & Georgiou (2015), yielding $H_1 = 0$. With that, the solution to (6) for quadratic $V(x) := \alpha\|x\|^2$ and $\sigma := 0$ can be obtained by solving the matrix ODE in (35) with $A = 0$, $B = \boldsymbol{I}_d$, $R = \frac{1}{2}\boldsymbol{I}_d$, and $Q = \alpha\boldsymbol{I}_d$, which admits an analytic solution in the form of hyperbolic tangent functions, similar to (19).

## D  EXPERIMENT DETAILS

### D.1  EXPERIMENT SETUP

Table 7: Hyperparameters of the `SplineOpt` (Alg. 3) for each task.

|  | Stunnel | Vneck | GMM | Lidar | AFHQ | Opinion |
|---|---|---|---|---|---|---|
| Number of control pts. $K$ | 30 | 30 | 30 | 30 | 8 | 30 |
| Number of gradient steps $M$ | 1000 | 3000 | 2000 | 200 | 100 | 700 |
| Number of samples $|i|$ | 4 | 4 | 4 | 4 | 4 | 4 |
| Optimizer | SGD | SGD | SGD | mSGD | Adam | SGD |

**Baselines.**  All experiments on DeepGSB are run with their official implementation[5] and default hyperparameters. We adopt the "actor-critic" parameterization as it generally yields better performance, despite requiring additional value networks. On the other hand, we implement DSBM by ourselves, as our GSBM can be made equivalent to DSBM by disabling the optimization of the CondSOC problem in (6) and returning the analytic solution of Brownian bridges instead. This allows us to more effectively ablate the algorithmic differences, ensuring that any performance gaps are attributed to the presence of $V_t$. All methods, including our GSBM, are implemented in PyTorch (Paszke et al., 2019).

**Network architectures.**  For crowd navigation (Sec. 4.1) and opinion depolarization (Sec. 4.3), we adopt the same architectures from DeepGSB, which consists of 4 to 5 residual blocks with sinusoidal time embedding. For the AFHQ task, we consider the U-Net (Ronneberger et al., 2015) architecture implemented by Dhariwal & Nichol (2021).[6] All networks are trained from scratch, without utilizing any pretrained checkpoint, and optimized with AdamW (Loshchilov & Hutter, 2019).

**Task-specific noise level ($\sigma$).**  For crowd navigation tasks with mean-field cost, we adopt $\sigma = \{1.0, 2.0\}$, whereas the opinion depolarization task uses $\sigma = 0.5$. These values are inherited from DeepGSB. On the other hand, we use $\sigma = 1$ and $0.5$ respectively for LiDAR and AFHQ tasks.

**GSBM hyperparameters.**  Table 7 summarizes the hyperparameters used in the spline optimization. By default, the generation processes are discretized into 1000 steps, except for the opinion depolarization task, where we follow DeepGSB setup and discretize into 300 steps.

**GSBM implementation (forward & backward scheme).**  In practice, we employ the same "forward and backward" scheme proposed in DSBM (Shi et al., 2023), parameterizing *two* drifts, one for the forward SDE and another for the backward. During odd epochs, we simulate the coupling (line 4 in Alg. 5) from the forward drift, solve the corresponding CondSOC problem (6), then match the resulting $p_t$ with the backward drift. Conversely, during even epochs, we follow the reverse process, matching the forward drift with the $p_t$ obtained from backward drift. The forward-backward alternating scheme generally improves the performance, as the forward drift always matches the ground-truth terminal distribution $p_1 = \nu$ (and vise versa for the backward drift).

---

[5] https://github.com/ghliu/DeepGSB, under Apache License.
[6] https://github.com/openai/guided-diffusion, under MIT License.

**Crowd navigation setup.** All three mean-field tasks—Stunnel, Vneck, and GMM—are adopted from DeepGSB, as shown in Fig. 11. We slightly modify the initial distribution of GMM to testify fully multi-model distributions. As for the mean-field interaction cost (12), we consider entropy cost for the Vneck task, and congestion cost for the Stunnel and GMM tasks. We adjust the multiplicative factors between $L_{\text{obstacle}}$ and $L_{\text{interact}}$ to ensure that noticeable changes occur when enabling mean-field interaction; see Table 8. These factors are much larger than the ones considered in DeepGSB ($\lambda_2 < 3$). In practice, we soften the obstacle cost for differentiability, similar to prior works (Ruthotto et al., 2020; Lin et al., 2021), and approximate $p_t(x) \approx \sum_{(x_0,x_1)} p_t(x|x_0,x_1)$ as the mixture of Gaussians.

**AFHQ setup.** As our state cost $V_t$ is defined via a latent space, we pretrain a $\beta$-VAE (Higgins et al., 2016) with $\beta = 0.1$. On the other hand, the spherical linear interpolation (Slerp; Shoemake (1985)) refers to constant-speed rotational motion along a great circle arc:

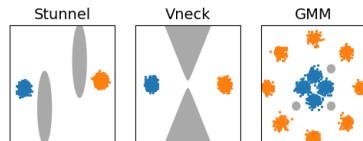

Figure 11: Initial and terminal distributions for each crowd navigation task with mean-field state cost. Obstacles are marked gray.

Table 8: Multiplicative factors of the state cost for each crowd navigation task: $V_t(x) = \lambda_{\text{obs}} L_{\text{obstacle}}(x) + \lambda_{\text{int}} L_{\text{interact}}(x; p_t)$.

|  | Stunnel | Vneck | GMM |
|---|---|---|---|
| $\lambda_{\text{obs}}$ | 1500 | 3000 | 1500 |
| $\lambda_{\text{int}}$ | 50 | 8 | 5 |

$$I_{\text{Slerp}}(t, z_0, z_1) := \frac{\sin((1-t)\,\Omega)}{\sin \Omega} z_0 + \frac{\sin(t\,\Omega)}{\sin \Omega} z_1, \qquad \Omega = \arccos(\langle z_0, z_1 \rangle). \tag{36}$$

### D.2  OPINION DEPOLARIZATION

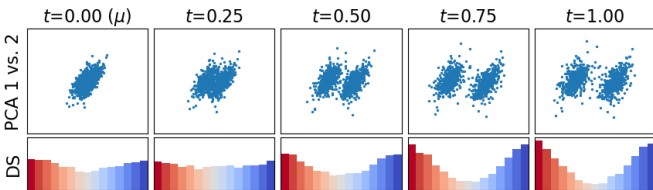

Figure 12: Simulation of the polarize drift (37) in $\mathbb{R}^{1000}$.

**Polarize drift in (15).** We use the same polarize drift from DeepGSB, based on the party model (Gaitonde et al., 2021). At each time step $t$, all agents receive the same random information $\xi_t \in \mathbb{R}^d$ sampled independently of $p_t$, then react to this information according to

$$f_{\text{polarize}}(x; p_t, \xi_t) := \mathbb{E}_{y \sim p_t} [a(x, y, \xi_t)\bar{y}], \quad a(x, y, \xi_t) := \begin{cases} 1 & \text{if } \text{sign}(\langle x, \xi_t \rangle) = \text{sign}(\langle y, \xi_t \rangle) \\ -1 & \text{otherwise} \end{cases},$$
$$\tag{37}$$

where $\bar{y} := y/\|y\|^{\frac{1}{2}}$ and $a(x, y, \xi_t)$—the agreement function—indicates whether the two opinions $x$ and $y$ agree on the information $\xi_t$. Hence, (37) suggests that the agents are inclined to be receptive to opinions they agree with while displaying antagonism towards opinions they disagree with. This is known to yield polarization, as shown in Figure 12.

**Directional similarity in Figures 7 and 12.** Directional similarity is a standard visualization for opinion modeling (Schweighofer et al., 2020) that counts the histogram of cosine angle between pairwise opinions. Hence, flatter directional similarity suggests less polarize opinion distribution.

### D.3  APPROXIMATING GEOMETRIC MANIFOLDS WITH LIDAR DATA

Since LiDAR data is a collection of point clouds in $[-5, 5]^3 \subset \mathbb{R}^3$, we use standard methods for treating it more as a Riemannian manifold. For every point in the ambient space, we define the projection operator by first taking a $k$-nearest neighbors and fitting a 2D tangent plane. Let $\mathcal{N}_k(x) = \{x_1, \ldots, x_k\}$ be the set of $k$-nearest neighbors for a query point $x \in \mathbb{R}^3$ in the ambient space. We

then fit a 2D plane through a moving least-squares approach (Levin, 1998; Wendland, 2004),

$$\underset{a,b,c}{\arg\min} \frac{1}{k} \sum_{i=1}^{k} w(x, x_i)(a x_i^{(x)} + b x_i^{(y)} + c - x_i^{(z)})^2 \tag{38}$$

where the superscripts denotes the $x, y, z$ coordinates and we use the weighting $w(x, x_i) = \exp\{-\|x - x_i\|/\tau\}$ with $\tau = 0.001$. We solve this through a pseudoinverse and obtain the approximate tangent plane $ax + by + c = z$. When $k \to \infty$, this tangent plane is smooth; however, we find that using $k = 20$ works sufficiently well in our experiments, and our GSBM algorithm is robust to the value of $k$. The projection operator $\pi(x)$ is then defined using the plane,

$$\pi(x) = x - \left( \frac{x^\top n + c}{\|n\|^2} \right) n, \qquad \text{where } n = \begin{bmatrix} a & b & -1 \end{bmatrix}^\top. \tag{39}$$

This projection operator is all we need to treat the LiDAR dataset as a manifold. Differentiating through $\pi$ will automatically project the gradient onto the tangent plane. This will ensure that optimization through the state cost $V_t$ will be appropriated projected onto the tangent plane, allowing us to optimize quantities such as the height of the trajectory over the manifold.

The exact state cost we use includes an additional boundary constraint:

$$V(x) = \lambda \Big[ \underbrace{\|\pi(x) - x\|^2}_{L_{\text{manifold}}} + \underbrace{\exp(\pi(x)^{(z)})}_{L_{\text{height}}} + \underbrace{\sum_{p \in \{x^{(x)}, x^{(y)}\}} \text{sigm}(p-5/0.1) + (1 - \text{sigm}(p+5/0.1))}_{L_{\text{boundary}}} \Big], \tag{40}$$

where "sigm" is the sigmoid function, used for relaxing the boundary constraint. The loss $L_{\text{boundary}}$ simply ensures we don't leave the area where the LiDAR data exists. We set $\lambda = 5000$. The gradient of $L_{\text{manifold}}(x)$ moves $x$ in a direction that is orthogonal to the tangent plane while the gradient of $L_{\text{height}}(x)$ moves $x$ along directions that lie on the tangent plane. If we only have $L_{\text{manifold}}$, then CondSOC essentially solves for geodesic paths and we recover an approximation of Riemannian Flow Matching (Chen & Lipman, 2023) (parameterized in the ambient space) when $\sigma \to 0$. The state cost $L_{\text{height}}$ depends on $\pi(x)$ as this ensures we travel down the mountain slope when optimizing for height. Other state costs can of course also be considered instead of $L_{\text{height}}$.

## D.4 ADDITIONAL EXPERIMENT RESULTS

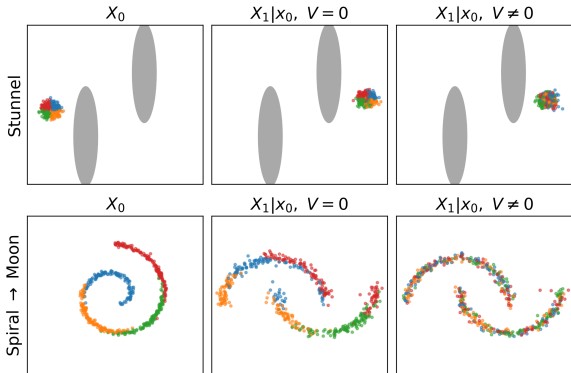

Figure 13: Illustration of how different couplings, $X_1|x_0$, emerge with nontrivial state costs $V(x)$.

**Nontrivial $V(x)$ induces different couplings.** Figure 13 illustrates how different couplings, $X_1|x_0$, emerge with nontrivial state costs $V(x)$. Specifically, we color different regions of the initial distribution (leftmost column) with different colors and track their pushforward maps. We consider the same $V(x)$, *i.e.*, obstacle and congestion costs, for Stunnel (top row) and adopt quadratic cost for Spiral → Moon (bottom row). In these two particular cases, having nontrivial $V(x)$ encourages stronger mixing, thereby yielding a different coupling (rightmost column) compared to the one induced without $V(x)$ (middle column).

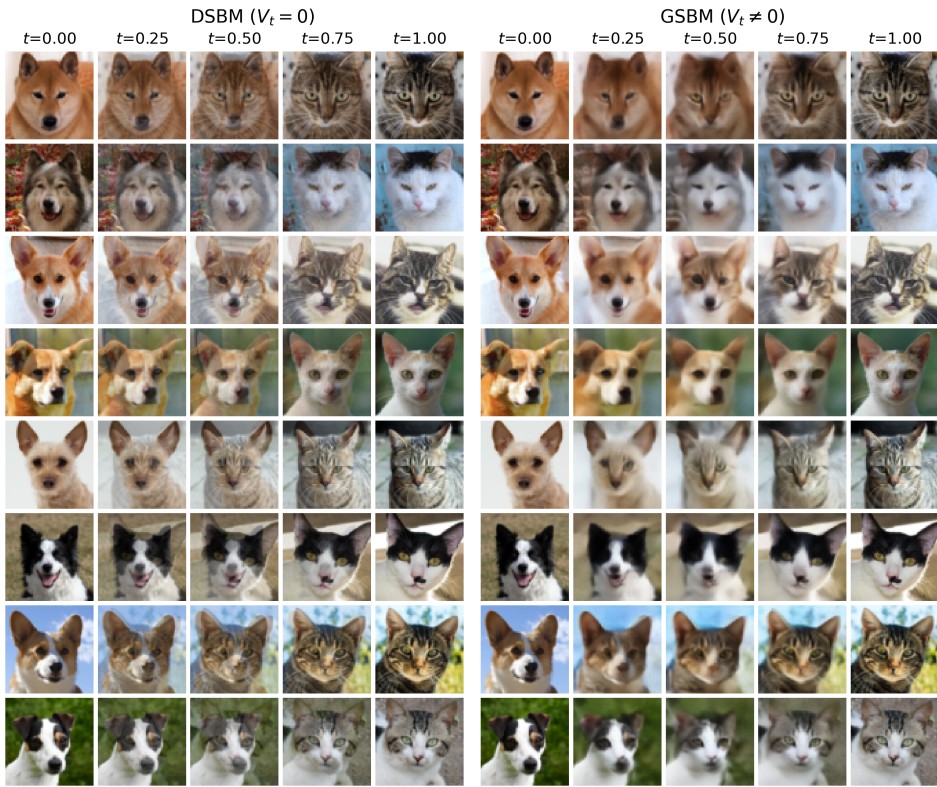

Figure 14: Additional comparison between DSBM (Shi et al., 2023) and our GSBM through the mean of $p_t(X_t|x_0, x_1)$ with randomly sampled $(x_0, x_1)$. While DSBM simply performs linear interpolation between $x_0$ and $x_1$, our GSBM *optimizes* w.r.t. (6) where $V_t$ is defined via a latent space, thereby exhibiting **more semantically meaningful interpolations**.

Table 9: Quantitative comparison between NLSB (Koshizuka & Sato, 2023) and our **GSBM**. Notice that our GSBM consistently ensures feasibility. In contrast, as NLSB approaches the GSB problem by transforming it into a stochastic optimal control (SOC) problem with a soft terminal cost (see Appendix A), it may overly emphasize on optimality at the expense of feasibility.

| | **Feasibility** $\mathcal{W}(p_1^\theta, \nu)$ | | | **Optimality** (3) | | |
| | Stunnel | Vneck | GMM | Stunnel | Vneck | GMM |
|---|---|---|---|---|---|---|
| NLSB (Koshizuka & Sato, 2023) | 30.54 | 0.02 | 67.76 | 207.06 | 147.85 | 4202.71 |
| **GSBM (ours)** | 0.03 | 0.01 | 4.13 | 460.88 | 155.53 | 229.12 |

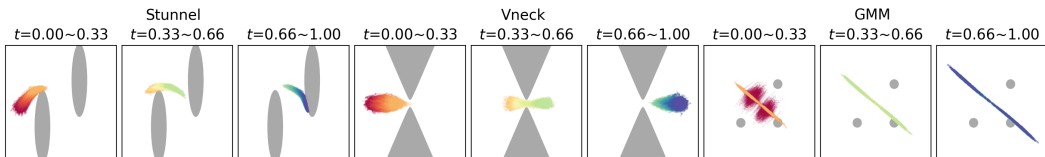

Figure 15: Performance of NLSB (Koshizuka & Sato, 2023) on the same crowd navigation tasks. Notice how NLSB was trapped in some local minima on Stunnel and completely failed on GMM.

**Additional interpolation results on AFHQ.** Figure 14 provides comparison results between DSBM (Shi et al., 2023) and our GSBM on $p_t(X_t|x_0, x_1$ with randomly sampled $(x_0, x_1)$.

**Additional comparisons on crowd navigation with mean-field cost.** Table 9 and Fig. 15 report the performance of NLSB[7] (Koshizuka & Sato, 2023)—an adjoint-based method for approximat-

---
[7]https://github.com/take-koshizuka/nlsb, under MIT License.

ing solutions to the same GSB problem (3). Figure 16 provides additional comparison between DeepGSB (Liu et al., 2022) and our GSBM. It should be obvious that our GSBM outperforms both methods.

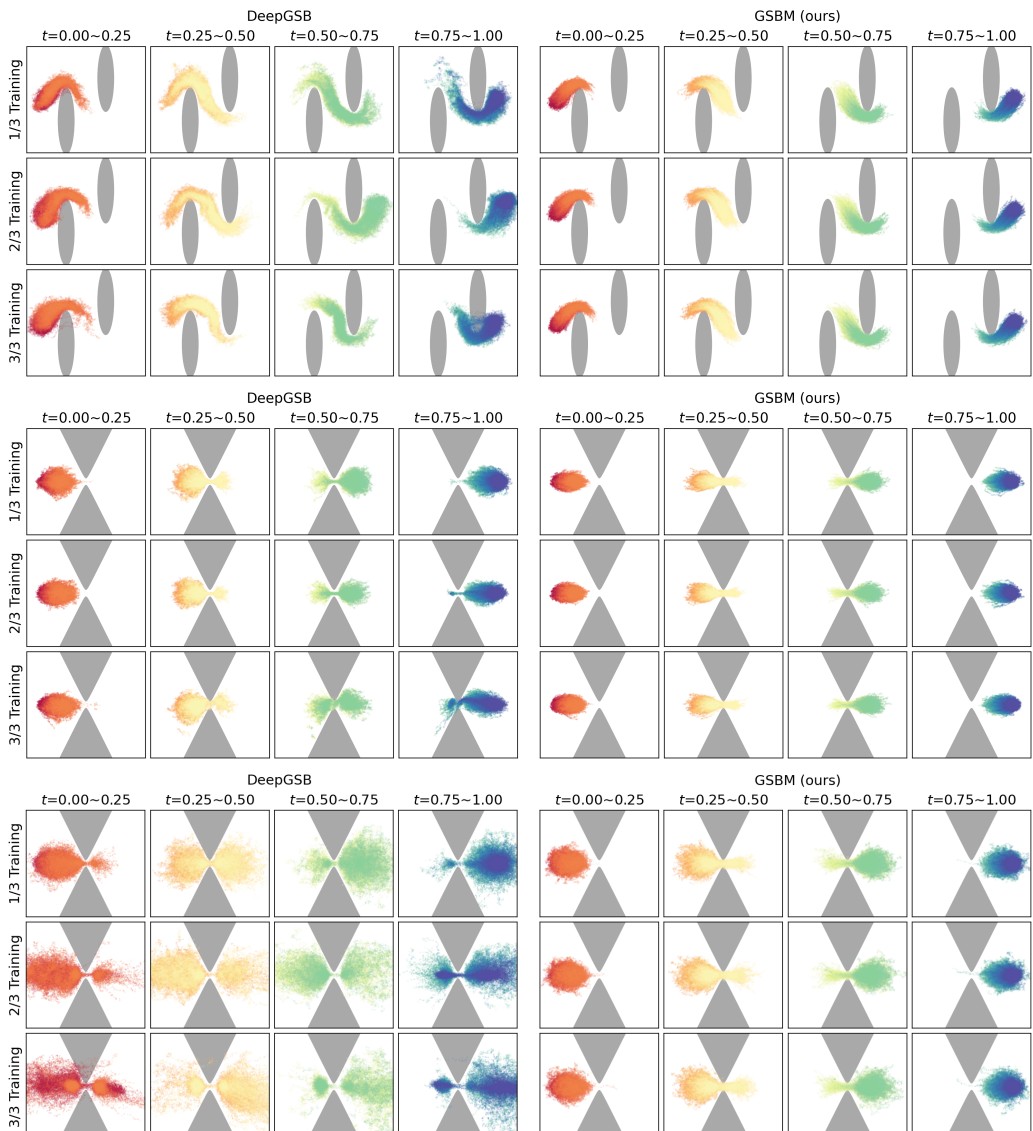

Figure 16: Additional comparison betwee DeepGSB (Liu et al., 2022) our GSBM on (*top to bottom*) Stunnel with $\sigma = 2.0$ and Vneck with $\sigma = \{1.0, 2.0\}$. Notice again how the training of DeepGSB exhibits relative instability and occasional divergence.

