# OpenReview forum: "Generalized Schrödinger Bridge Matching"
_ICLR.cc/2024/Conference — ICLR 2024 poster_

### Official Review · Reviewer_fcZ9 · 2023-10-30

**Soundness:** 2 fair
**Presentation:** 3 good
**Contribution:** 3 good
**Rating:** 8
**Confidence:** 4

**Summary:**

This paper propose Generalized Schrodinger Bridge Matching (GSBM) that can be view as solving conditional stochastic optimal control, for which efficient variational approximations can be used, and further debiased with the aid of path integral theory.  Compared to other methods for solving GSB problems, GSBM algorithm preserves a feasible transport map between the boundary distributions throughout training, thereby enabling stable convergence and significantly improved scalability.

**Strengths:**

The paper has rigorous logic, clear expression and abundant experimental details. This work broadens the scope of application of Schrodinger bridge matching (SBM) algorithm and converts the existing optimal transport mapping and SBM methods into the special case of variational formulas. In addition, the feasibility conditions are met at any time during the entire training process, which is a big difference from the previous methods. Overall, this work opens up new algorithmic opportunities for training diffusion models with task-specific optimal structures.

**Weaknesses:**

The Generalized Schodinger Bridge Matching (GSBM) proposed in this paper is not applicable to any form of V_{t}, but requires the differentiability of V_{t} and relies on quadratic control costs to establish its convergence analysis. Although it is a significant improvement over previous GSB solvers, the Generalized Schodinger Bridge Matching (GSBM) method is not applicable to any form of V_{t}. But differentiability is still a necessary condition.

**Questions:**

Does (6) have an analytic solution only if V_{t} is quadratic and \sigma>0?

---

> ### Author Response · Authors · 2023-11-17
> **Author Response to Reviewer fcZ9**
>
> **1. Remarks on global convergence**
>
> - We kindly note the significance of monotonic non-increase in the training objective, which implies training stability, particularly in the context of learning-based methods. This is the same type of analysis done for standard alternating optimization algorithms such as Expectation-maximization (EM) which only has this type of (local) convergence guarantees. Nevertheless, we do agree that convergence analysis could be improved for certain special cases of the state cost. For example, when the state cost $V(x)$ penalizes the deviation from a compact manifold (similar to the manifold cost defined in Eq 13), GSB should degenerate to Riemannian SB [1] given a sufficiently large $V$, and from which existing convergence analysis could be adopted.
>
> - A more rigorous statement could be attained by re-examining existing proofs of global convergence for DSBM [2], _i.e._, when $V=0$. Below, we provide a proof sketch to illustrate the concept. Specifically, the global convergence of DSBM hinges on two steps: first, casting the l2 norm $\mathbb{E}[\int\|u_t\|^2]dt$ as the KL divergence between a controlled process and a reference process $\mathbb{Q}$, which, in such cases, is simply a Brownian motion; and secondly, showing that $\mathbb{Q}$ satisfies regularity conditions (see Prop 5 in [2]) such that the fixed point of the learning algorithm coincides with the unique solution.
>
> - Now, it has been shown that the objective in Eq 3 can also formulated as a KL divergence [3], except that its reference process is now reweighed with the state cost: $\tilde{\mathbb{Q}} \propto \mathbb{Q} \exp(\int_0^1 V_t(X_t) dt)$. Notice that when $V:=0$, $\tilde{\mathbb{Q}}$ degenerates to $\mathbb{Q}$ as expected. This suggests that if we can identify a family of $V$ such that their corresponding $\tilde{\mathbb{Q}}$ obey similar regularity conditions, convergence analytic from [2] may be carried over for our GSBM. This is an interesting theoretical question albeit requiring additional works, nevertheless we thank the reviewer for raising these comments.
>
> ---
>
> **2. Differentiability of the state cost $V$**
>
> - We note that the differentiability of $V$ is required _only_ when adopting spline parameterization (Alg 3) for solving the conditional stochastic optimal control problem (CondSOC; Eq 6). That being said, one may consider alternative solvers to CondSOC solver that, albeit less efficient, provide an approximate solution to Eq 6 yet without having to assume differentiability of $V$. This includes _(i)_ solving Eq 6 directly with path integral as in our Prop 4, _(ii)_ enforcing the HJB PDE in Eq 18 with an PINN loss [4,5], or _(iii)_ expanding Eq 18 with nonlinear Feynman Kac lemma and then applying temporal difference loss, as initially proposed in [6].
>
> - When $V$ is differentiable and known in priori---which is a realistic setup for a wide range of general distribution matching problems as demonstrated in our experiments, gradient-based solvers generally enjoy superior scalability and computational complexity, hence leveraging by our GSBM.
> ---
>
> **3. Other cases when CondSOC (Eq 6) has analytic solutions**
>
> - We've identified two other cases, apart from the configuration mentioned by the reviewer, when Eq 6 has analytic solutions. First, as briefly sketched in 1., for a sufficiently large $V$ that penalizes deviation from a compact manifold and $\sigma :=0$, Eq 6 accounts to solving the geodesic path, which admits analytic solutions for manifolds such as sphere, tori, hyperbolic ball, and SPD matrices. Secondly, for quadratic $V$ and $\sigma :=0$, Eq 6 reduces to a linear quadratic regulator with two end-point constraints. The solution relates to a Lyapunov differential equation that admits analytic solution. We detail the derivation in Appendix E (Page 25) for completeness. For general state costs considered in Sec 3.2, solving Eq 6 necessitates solving its necessary condition in Eq 18 (see Page 16). Hence, as long as Eq 18 has analytic solution for some given $V$, so does Eq 6. This is an interesting question, and we thank the reviewer for raising it.
>
> ---
>
> [1] Riemannian diffusion SB (https://arxiv.org/abs/2207.03024)
> [2] Diffusion SB matching (https://arxiv.org/abs/2303.16852)
> [3] Linearly-solvable MDP (https://homes.cs.washington.edu/~todorov/papers/TodorovNIPS06.pdf)
> [4] Physics-informed neural networks (https://arxiv.org/abs/1711.10561)
> [5] Transport, variational inference & diffusions (https://arxiv.org/abs/2307.01050v1)
> [6] Deep generalized SB (https://arxiv.org/abs/2209.09893)

---

> ### Comment · Area_Chair_7Gp2 · 2023-11-20
> **Respond to authors' rebuttal**
>
> Please, confirm that you have read the author's response and the other reviewers' comments and indicate if you are willing to revise your rating.

---

### Official Review · Reviewer_AA2i · 2023-11-01

**Soundness:** 3 good
**Presentation:** 3 good
**Contribution:** 3 good
**Rating:** 8
**Confidence:** 2

**Summary:**

The paper considers a novel method for the generalized Schrodinger bridge method, which allows for incorporation of a control cost into the interpolation. Their method works by alternating between optimizing the drift and the marginals, with the second done via a tie to conditional stochastic optimal control. In this second step, the optimal probability path between two fixed endpoint marginal samples is approximated with a Gaussian spline (splines separately for the time-varying means and standard deviations). The method is tested and compared primarily against DeepGSB in crowd navigation, image interpolation, and opinion depolarization.

**Strengths:**

The paper presents a method which improves upon previous methods by ensuring guaranteed feasibility and stronger performance in high dimension. The method also seems to clearly outperform comparable methods in the various application settings.

**Weaknesses:**

* The convergence guarantees obtained are merely local, and do not guarantee convergence to a global optimum.
* As noted in the conclusion, the method requires a differentiable state cost.

**Questions:**

In relation to the first item above:
1. Can you prove convergence to a global optimum for a particular class of state cost V?
2. If not, do you have any commentary on other strategies for initialization of the spline optimization?

---

> ### Author Response · Authors · 2023-11-17
> **Author Response to Reviewer AA2i (part 1/2)**
>
> **1. Remarks on global convergence**
>
> - We kindly note the significance of monotonic non-increase in the training objective, which implies training stability, particularly in the context of learning-based methods. This is the same type of analysis done for standard alternating optimization algorithms such as Expectation-maximization (EM) which only has this type of (local) convergence guarantees. Nevertheless, we do agree that convergence analysis could be improved for certain special cases of the state cost. For example, when the state cost $V(x)$ penalizes the deviation from a compact manifold (similar to the manifold cost defined in Eq 13), GSB should degenerate to Riemannian SB [1] given a sufficiently large $V$, and from which existing convergence analysis could be adopted.
>
> - A more rigorous statement could be attained by re-examining existing proofs of global convergence for DSBM [2], _i.e._, when $V=0$. Below, we provide a proof sketch to illustrate the concept. Specifically, the global convergence of DSBM hinges on two steps: first, casting the l2 norm $\mathbb{E}[\int\|u_t\|^2]dt$ as the KL divergence between a controlled process and a reference process $\mathbb{Q}$, which, in such cases, is simply a Brownian motion; and secondly, showing that $\mathbb{Q}$ satisfies regularity conditions (see Prop 5 in [2]) such that the fixed point of the learning algorithm coincides with the unique solution.
>
> - Now, it has been shown that the objective in Eq 3 can also formulated as a KL divergence [3], except that its reference process is now reweighed with the state cost: $\tilde{\mathbb{Q}} \propto \mathbb{Q} \exp(\int_0^1 V_t(X_t) dt)$. Notice that when $V:=0$, $\tilde{\mathbb{Q}}$ degenerates to $\mathbb{Q}$ as expected. This suggests that if we can identify a family of $V$ such that their corresponding $\tilde{\mathbb{Q}}$ obey similar regularity conditions, convergence analytic from [2] may be carried over for our GSBM. This is an interesting theoretical question albeit requiring additional works, nevertheless we thank the reviewer for raising these comments.
>
> ---
>
> **2. Differentiability of the state cost $V$**
>
> - We note that the differentiability of $V$ is required _only_ when adopting spline parameterization (Alg 3) for solving the conditional stochastic optimal control problem (CondSOC; Eq 6). That being said, one may consider alternative solvers to CondSOC solver that, albeit less efficient, provide an approximate solution to Eq 6 yet without having to assume differentiability of $V$. This includes _(i)_ solving Eq 6 directly with path integral as in our Prop 4, _(ii)_ enforcing the HJB PDE in Eq 18 with an PINN loss [4,5], or _(iii)_ expanding Eq 18 with nonlinear Feynman Kac lemma and then applying temporal difference loss, as initially proposed in [6].
>
> - When $V$ is differentiable and known in priori---which is a realistic setup for a wide range of general distribution matching problems as demonstrated in our experiments, gradient-based solvers generally enjoy superior scalability and computational complexity, hence leveraging by our GSBM.
>
> ---
>
> [1] Riemannian diffusion SB (https://arxiv.org/abs/2207.03024)
> [2] Diffusion SB matching (https://arxiv.org/abs/2303.16852)
> [3] Linearly-solvable MDP (https://homes.cs.washington.edu/~todorov/papers/TodorovNIPS06.pdf)
> [4] Physics-informed neural networks (https://arxiv.org/abs/1711.10561)
> [5] Transport, variational inference & diffusions (https://arxiv.org/abs/2307.01050v1)
> [6] Deep generalized SB (https://arxiv.org/abs/2209.09893)

---

> ### Author Response · Authors · 2023-11-17
> **Author Response to Reviewer AA2i (part 2/2)**
>
> **3. Strategy for initialization of spline optimization (Alg 3)**
>
> - In practice, we find that the spline optimization (Alg 3) is quite robust to hyper-parameters. In the two tables below, we report the optimized objective values (Eq 6a) by initializing Alg 3 with variations in how the controlled points are distributed across time steps (regular (first table) vs irregular (second table)) and different numbers of controlled points for parameterizing the mean $\mu_t$ and standard deviation $\gamma_t$ splines (see Eq 9). The initial objective value is much larger relatively, around 18000. Hence, it is evident that across all configurations, the optimization converges to similar values. Furthermore, we observe stable convergence in all cases. We highlight these results as a consequence of how our GSBM decomposes the GSB problem (see Sec 3.1), which yields a variational problem (Prop 2 & Eq 6) that can be optimized more simply and steadily (since it is constrained by two end-points instead of two distributions).
>
>   |  | Num of controlled points in $\mu_t$ = 5 | 10 | 15 | 20 |
>   |---|---|---:|---:|---:|
>   | Num of controlled points in $\gamma_t$ = 10 | 448.71 | 449.80 | 439.11 | 439.92 |
>   | 20 | 454.71 | 452.88 | 439.81 | 439.92 |
>   | 30 | 459.50 | 456.13 | 444.10 | 443.14 |
>   | 40 | 462.24 | 459.23 | 446.05 | 446.34 |
>
>   |  | 5 | 10 | 15 | 20 |
>   |---|---:|---:|---:|---:|
>   | 10 | 449.37 | 449.74 | 439.39 | 440.04 |
>   | 20 | 449.22 | 454.84 | 440.91 | 441.28 |
>   | 30 | 455.23 | 454.64 | 443.15 | 443.64 |
>   | 40 | 464.27 | 454.97 | 446.19 | 445.31 |

---

> ### Comment · Area_Chair_7Gp2 · 2023-11-20
> **Respond to authors' rebuttal**
>
> Please, confirm that you have read the author's response and the other reviewers' comments and indicate if you are willing to revise your rating.

---

> > ### Comment · Reviewer_AA2i · 2023-11-22
> > **Response acknowledged**
> >
> > I thank the authors for their extensive response, and am happy to keep my score where it is.

---

### Official Review · Reviewer_cBUM · 2023-11-01

**Soundness:** 3 good
**Presentation:** 2 fair
**Contribution:** 3 good
**Rating:** 6
**Confidence:** 5

**Summary:**

This paper focuses on the problem of controlled measure transport. Given a time dependent density $p_t(x)$, how do you adapt $p_t(x)$ with respect to some potential function $V_t(x)$ so that it arrives at a target density $p_{T}(x)$ and does so in some measure of least cost (or as they refer to it, energy).

They rely on recent distribution matching algorithms to do this, writing the discovery of some vector field $s_t(x)$ and the least cost set of $p_t(x)$ that correspond in Fokker-Planck sense to this $s_t(x)$ as the solution to two constrained minimization problems.

The reviewer interprets the "generalized" in the title to refer to the fact that the paper moves away from the notion of just connecting two densities, but also opens the avenue for doing this e.g. with obstacles (a la optimal control problems).

They demonstrate their method e.g. on toy blockade examples and with the effects of stochasticity on the energy minimization, as well on image-to-image translation examples and a cool LiDAR surface example.

**Strengths:**

The authors exposit their work quite well (with adjustments needed mentioned in the weaknesses and questions section). The experimental section is thorough and highlights the myriad applications of the perspectives presented in the paper. With the proper contextualization, the method presented is interesting and falls in line well with a series of other related works, highlighting the benefits of controllable paths (for the limited set of potentials the consider).

**Weaknesses:**

While the experiments are cool and thorough, and while the framing of the optimization problem is coherent, the paper diminishes itself by its framing. *This is an addressable issue* and the reviewer thinks it can be straightforwardly, especially during a rebuttal period, as it's not an experimental thing. The framing issue is that it oversells itself in a way that misleads the reader.

I think a good principle of academia (that we can all reasonably agree with) is to only assert what we can reasonably back-up and to not mislead. This paper does not provide a numerical solution to optimal transport or the schrodinger bridge. In the introduction, it casts OT and Schrodinger Bridge as edge-cases of their variational problem. *However, they do not provide an actual solution to that variational problem*. The statement of this problem has been well known. The **application** of it in the context of generative modeling and dynamical systems is compelling (even if approximate! you don't need to solve OT/SB to do cool control!) but it does not need to be couched as if it is solving those problems. It is still compelling to say "we came up with an approximate way to do stochastic optimal control in the context of deep generative modeling" without saying "we have provided a variational method for solving optimal transport and SB."

The issue is that the constraint (6b) is actually quite hard to enforce. The Gaussian approximation is fine. But the authors should really be careful in stating what it is they are doing and how it relates to the problem they are seeking to solve, because gaussian paths are not going to do it. The paper that this one primarily cites (Flow Matching for Generative Modeling) seems to do this too -- in which it specifies an "optimal transport path" that isn't actually providing a means to solve OT.

The reason the reviewer stresses this point is two-fold: the first is that mathematicians have spent many years carefully characterizing the hardness of these problems and approximate solutions without overstating them. The second is that doing this induces confusion to new readers who come into the field. This is a damaging effect, it muddles points and slows down new members, and one the reviewer has already had to deal with from students misinterpreting the claims of OT in these types of papers.

This can be addressed by simply toning down some of claims, and including these caveats, e.g. in the bullet points on page two summarizing the papers contributions.

**References**

There are some necessary adjustments to the citations needed, but that can be cleared up. Here are some things that should be addressed:

- At the beginning of section 3.1: Neither [Liu 2023b] nor [Peluchetti 2022] propose any sort of entropic optimal transport. Perhaps there was a typo in the citep. While Liu 2023b proposes a procedure for OT (when the iterating field is a gradient field), nothing about it is entropic.
- There seems to be some missing citations with regards to the matching algorithms. On page one following the line "a mixture of tractable conditional probability paths (Ho, Song, Lipman)" should include [1,2,3] listed below. These should likewise be included Bridge and Flow matching (explicit) section. There is also related follow-up work by Tong et al listed as [4]
- The Schrodinger Bridge problem in the introduction should not be attributed to the machine learners. It's been studied for decades. The original citation is provided in [5], and a follow-up is provided in [6].


**Other adjustments**
- The discussion about feasibility seems not so useful. The statement of the problem the paper wants to solve is that the solution is feasible (it's chicken-egg). That's the point of a constraint in an optimization problem. If the one paper the authors want to compare this work to (the Generalized Schrodinger Bridge one) fails to meet this, it doesn't seem worth taking the reader down that line when the problem statement assumes it. Table 1 could be in an appendix :)
- There are some minor things throughout the text to adjust, which are semantically inexact. For example: in the line in the "Implicit action matching" paragraph:  "...then the unique gradient field $\nabla s^\theta_t(X_t)$ that matches $p_t$ can be obtained by minimizing...". The noun comparison is wrong here. $\nabla s^\theta_t(Xt)$ does not match a probability density. The learning problem is to find the $\nabla s^\theta_t(X_t)$ for which the probability density *of the pushforward* from the initial density $p_0$ to $\tilde p_t$ by this $s_t^\theta$ matches $p_t$. Densities match densities.
- The authors should adjust the first paragraph of the second page to highlight that optimal transport as a field does consider a wide variety of other costs besides the $\mathcal W-2$ cost. There are e.g. repulsive costs, e.g. in the study of quantum dynamics, etc. The notion of optimality still holds.
- Lastly, are the authors sure that a reasonably different coupling $p(x_0, x_1)$ emerges at the minimizer? Neither set of experiments seems to suggest this. While it is true in the image-to-image translation, that the intermediate density $p_t(x)$ has a more semantically meaningful translation, the marginal endpoints (the $x_0$ associated with $x_1$) for both algorithms look pretty much identical. Moreover, Figure 1 highlights an equivalent picture. The initial and final empirical densities (light green and dark green) are identical even after learning.


**The reviewer is more than happy to improve their score if the framing, citations, and "other adjustments"**. As of now I would ideally mark this as a 4/10 and not a 5/10, but that isn't an option any more it seems.

[1] Flow Straight and Fast: Learning to Generate and Transfer Data with Rectified Flow. Xingchao Liu, Chengyue Gong, Qiang Liu, Sept 2022.

[2] Building Normalizing Flows with Stochastic Interpolants. Michael S. Albergo and Eric Vanden-Eijnden, Sept 2022.

[3] Stochastic Interpolants: A Unifying Framework for Flows and Diffusions. Michael S. Albergo, Nicholas M. Boffi, Eric Vanden-Eijnden, March 2023.

[4] Improving and generalizing flow-based generative models with minibatch optimal transport. *Alexander Tong, Nikolay Malkin, Guillaume Huguet, Yanlei Zhang, Jarrid Rector-Brooks, Kilian Fatras, Guy Wolf, Yoshua Bengio*, July 2023.

[5] Schr¨odinger, E., “Uber die Umkehrung der Naturgesetze,” ¨ Sitzungsberichte der Preuss Akad. Wissen. Phys. Math. Klasse, Sonderausgabe, Vol. IX, 1931, pp. 144–153.

[6] R. Fortet, R´esolution d’un syst`eme d’equations de M. Schr¨odinger, J. Math. Pure Appl. IX (1940), 83-105.

**Questions:**

- A comment rather than a question: For an exciting read on the problem, the authors should see [6] , it's quite fun.
- Can the authors come up with a problem for which conditional density is actually far from Gaussian, or show that any of their examples currently are far from Gaussian?
- Can the authors try to quantify the "semantically meaningful" coupling?





[6] Stochastic control liaisons: Richard Sinkhorn meets Gaspard Monge on a Schroedinger bridge, *Yongxin Chen, Tryphon T. Georgiou, Michele Pavon*, 2020.

---

> ### Author Response · Authors · 2023-11-17
> **Author Response to Reviewer cBUM (part 1/2)**
>
> **1. Summary on the new presentation**
>
> **1.1 Clarification between GSB problem and GSBM algorithm**
>
> - We first thank the reviewer for the valuable and detailed comments. We agree that the relation between the practical instantiation of GSBM algorithm (Alg 5) and GSB problem itself (Eq 3) can be stated more clearly and could've led to unintentional confusion. We value academic papers as pivotal means for communication and place high standards on clear and rigorous expression (as recognized by Reviewer 7Jx7, fcZ9). Misleading readers is the last thing we wish to inadvertently propagate. We agree with the reviewer that some of our claims, as explained in the discussion below, are only based on empirical findings, not theoretical guarantees, and should be distinguished as such. We aim to clarify this in the paper.
>
> - To restate the relationship between GSBM and GSB, and as the reviewer has already recognized: The GSBM algorithm provides an _approximate solution_ to the GSB problem. The problem itself has great potential, albeit relatively unexplored, for tackling more general distribution matching tasks and, as shown in this work, benefits equally from recent advances in diffusion models. Meanwhile, the contribution of the proposed GSBM algorithm is to provide an approximate solution to GSB that, compared to prior deep learning-based methods, scales much more favorably to higher dimensional and more realistic setups considered in machine learning (such as images and point cloud data).
>
> - The "approximation" of GSBM to GSB comes from two places. First is the Gaussian path approximation to Eq 6 for tractability. The second approximation, which is shared across _all_ matching algorithms, is the fact that it may be practically unlikely to reach the unique minimizer given prescribed $p_t$ (_i.e._, line 1 in Alg 5 may not fully converge). This, as pointed out by the reviewer, could lead to violation of the boundary constraint. We note that we set the Gaussian path to satisfy the boundary constraints in Eq 6b by construction, so we took the reviewer's comment to mean that we cannot satisfy the distributional boundary constraints in Eq 2, which we agree with. In practice, we still found our alternating optimization approach to perform much more favorably compared to some existing deep learning-based approaches (_e.g._, DeepGSB [1], NLSB [2]) in approximating solutions to the same problem.
>
> - Nevertheless, we agree that both aforementioned approximations could be stated more clearly for improved clarification. In the revision, we revise abstract, Sec 1, Sec 3.2, and conclusion. Notable changes are marked in dark red. We aim to re-emphasize that modern learning algorithms (_e.g._, DeepGSB [1], DSBM [4]) provide _approximate_ solutions to GSB/SB problems, and our GSBM algorithm is no exception. Compared to prior methods [1,2,3], approximate solutions provided by "matching" algorithms (like our GSBM) remain much closer to the feasible set throughout training, and the conditions for them to preserve the exact marginals (thereby obeying Eq 6b) are noted in Footnote 2. However, when discussing such comparisons, we will make it more clear that this is based on intuitions regarding the algorithmic framework and our empirical findings, rather than a theoretical guarantee. Due to page limit, Table 1 is moved to Appendix A.
>
> - Finally, we note that the matching algorithms in this work can be divided into 3 classes. Algorithms such as Bridge Matching, Action Matching, and, as brought up by the reviewer, Flow Matching, do not provide a means to OT/SB (as stated in our introduction, we categorize such methods as those which _prescribe_ the probability path and does not optimize it). In contrast, algorithms such as Rectified Flow and DSBM aim to solve---approximately---OT/EOT with l2 cost via an additional outer loop similar to lines 4-5 in Alg 5. Our GSBM offers an intuitive explanation of such the second class of "optimality-enhanced" methods within the context of an alternating optimization and generalizes them beyond kinetic energy to include nontrivial state costs. We summarize the above discussions in Fig 9 on Page 14.
>
> ---
>
> **1.2 Reference & other adjustments**
>
> - In the revision, the original references on SB are added on Page 1. Other suggested references are added on Pages 1 & 3 and distributed in the current way due to page limit, as we're not allowed to add a new page to the main paper during rebuttal this year. We kindly note that [Liu 2022] & [Albergo 2023] were already cited in the initial submission but indeed should have been included in the reviewer's suggested places. The citep typo in Sec 3.1 is now fixed.
>
> - We restate the matching interpretation in Sec 2 & 3.1, and add additional comments on OT with general costs on Page 2. We thank the reviewer for the meticulous reading in strengthening the paper's rigor.
>
> ---
>
> [1] Deep generalized SB
> [2] Neural Lagrangian SB
> [3] Diffusion SB
> [4] Diffusion SB matching

---

> ### Author Response · Authors · 2023-11-17
> **Author Response to Reviewer cBUM (part 2/2)**
>
> **1.3 Discussion on feasibility**
>
> - We do agree that feasibility is often traded off with optimality, and approximate solutions by learning-based methods may inevitably encounter mismatches with feasibility constraints. However, we believe that the approach employed in learning algorithms to address feasibility are crucial factors that influence scalability, particularly when learning-based methods are applied to tackle the _same_ problem. On one hand, methods that simply relax the hard constraints (_e.g._, [2]) either necessitate running adjoint methods, thereby greatly hindering scalability, and require designing some terminal cost (_e.g._, through adversarial training [5]), which complicates the training & tuning (see our discussion on this in Appendix A). On the other hand, methods like [1,3] are able to deal with hard constraints but still face scalability issues, as feasibility is enforced by iterative projections in path space, which requires caching trajectories at all times. Compared to the aforementioned algorithms, _matching_ algorithms also tackle hard constraints yet are much more favorable in their practicality.
>
> - Given the aforementioned reasoning, we compare GSBM to DeepGSB [1] in Table 1, which is now moved to Table 6 in Appendix A, as both methods aim to approximate the solutions to GSB (Eq 3) _without_ relaxing Eq 2 (_i.e._, when there exists data, we should make use of it). On top of this same basis, this table aims to characterize their algorithmic distinction, essentially between matching vs non-matching methods. In the revision, we update the table and add additional comments in Sec 1 to avoid future confusion. We thank the reviewer for the comments.
>
> ---
>
> **2. Non-Gaussian conditional path**
>
> - An instance where the conditional density $p_{t|0,1}$ may deviate from Gaussian occurs when the solution to Eq 6 exhibits multi-modality. In such a case, Alg 3 converges to only one of the modalities. This, however, could potentially be addressed by extending Alg 3 with a mixture of Gaussian paths or simulating model coupling with multiple seeds (provided multi-modality is learned in Stage 1). We leave them for exploration in future work. We also note that the emergence of multi-modal solutions depends on the critical value of $\sigma$, as demonstrated in Fig 8. Throughout our experiments, we found that despite approximating the conditional density $p_{t|0,1}$ as uni-modal, the density $p_t$, obtained by marginalizing over $p_{0,1}$, can still exhibit multi-modality.
>
> ---
>
> **3. Clarification on coupling**
>
> - We first emphasize that Fig 1 is _NOT_ a learning result, but rather an illustration of how Alg 3 functions. As Alg 3 optimizes only the intermediate samples $X_t$ given _fixed_ coupling $(x_0,x_1)$, the initial and final empirical densities are identical by construction. More precisely, Fig 1 was generated as follows: _(i)_ sample $(x_0,x_1) \sim \mu \otimes \nu$, _(ii)_ initialize the mean & std of $X_t|x_0, x_1$ with Brownian bridge, _(iii)_ call Alg 3 for optimized $X_t^\prime|x_0, x_1$, and, finally, _(iv)_ plot $X_t|x_0, x_1$ and $X_t^\prime|x_0, x_1$ given the _same_ set of $(x_0,x_1)$.
>
> - In practice, we do observe different couplings with nontrivial state costs $V(x)$. This is demonstrated in Fig 16 in Appendix E, where enabling $V(x)$ encourages stronger mixing between samples, resulting in different couplings compared to those induced without $V(x)$. As for the image couplings (Fig 5, two columns annotated with "Epoch 8"), although they may appear similar at first glance, certain couplings exhibit variations in color (_e.g._, first sample) or distinct semantics (eyes in the third sample). We acknowledge that further improvements could be achieved by designing a different $V$ (which itself is an intriguing open question beyond the scope of this work) or considering a larger dataset.
>
> - For unsupervised image translation, most prior works rely on qualitative comparison as in our Fig 5,6,15. Quantitative metrics are less common, and we are only aware of human evaluation on online crowd-sourcing platform [6]. This is infeasible due to the limited rebuttal period. Nevertheless, in the table below, we attempt to quantify semantically meaningful "interpolations" via FID values (lower the better). Specifically, we provide the FID scores between our GSBM vs DSBM [4] at $t$ = {0.25, 0.5, 0.75}, considering the statistics of the dog, combined dog + cat, and cat datasets, respectively. The result indicates that our GSBM is much closer to the data distributions (with ground-truth semantics) compared to DSBM.
>
>   |  | t = 0.25 | 0.5 | 0.75 |
>   |---|---|---|---|
>   | DSBM [4] | 196.18 | 248.23 | 259.55 |
>   | **GSBM (ours)**  | 89.66 | 122.85 | 131.03 |
>
> ---
>
> [5] Alternating population & control NNs to solve high-dim MFG (https://arxiv.org/abs/2002.10113)
> [6] Guided image synthesis & editing with SDEs (https://arxiv.org/abs/2108.01073)

---

> > ### Comment · Reviewer_cBUM · 2023-11-21
> > **Thanks for your clarifications in the text**
> >
> > Thanks kindly to the authors for addressing some of my framing remarks regarding OT and the discussion regarding the approximation with gaussian conditional paths. I hope you also feel that this better contextualizes this in the various history and objectives of OT writ large.
> >
> > I am happy to up my score, but I must say I'm still a bit unconvinced by the coupling discussion. It is hard to say what the source of variation is in the final image -- it could be from a new coupling, or it could just be that what your algorithm has done is chosen another path in the space of measures that changes the *learnability* and results in the different practical output. It would more compelling if you had the analytic solution to such a set, say, a Gaussian mixture model for which the vector fields may be potentially be analytically available for which you can show that the *minimizer* gives a new coupling.
> >
> > Thanks again.

---

> > > ### Author Response · Authors · 2023-11-22
> > > **Author Response to Reviewer cBUM**
> > >
> > > We thank the reviewer for the reply and greatly appreciate the decision to raise the score.
> > >
> > > We agree that improving the learnability by constructing a different/better probability path is an interesting perspective provided by our GSBM. The revision has been updated accordingly, with modifications made to the caption of Fig 5 to reflect the current discussions. We thank the reviewer again for the comments.

---

> ### Comment · Area_Chair_7Gp2 · 2023-11-20
> **Respond to authors' rebuttal**
>
> Please, confirm that you have read the author's response and the other reviewers' comments and indicate if you are willing to revise your rating.

---

### Official Review · Reviewer_7Jx7 · 2023-11-06

**Soundness:** 3 good
**Presentation:** 3 good
**Contribution:** 3 good
**Rating:** 6
**Confidence:** 4

**Summary:**

The authors consider so-called generalised Schrödinger Bridge Matching problem: unlike standard SBM, which can be formulated as an optimisation problem of a drift of some diffusion, the corresponding optimisation problem in case of GSBM involved an additional state cost. The authors proposed an iterative algorithm to solve the problem. One of interesting tricks, proposed to increase computational efficiency of the solution, is a spline-based  approximation. The authors verified the algorithm for GSBM problem on a number of toy and practical tasks.

**Strengths:**

- the authors consider an import problem statement, which is a very natural generalisation of the SBM problem

- the proposed approach is theoretically sound

- the authors test their approach on a diverse set of problems

- the text is easy to follow

**Weaknesses:**

- the authors did not investigate computational limits of the proposed approach

- it seems that the proposed approach does not work in case of image data, only if we consider some latent space like in sec. 4.2. Any ideas how we can improve here?

**Questions:**

- how do computational efficiency and accuracy scale w.r.t. dimensionality? sample size?

- how do specific strategies for placement of spline control points influence accuracy (see sec. 3.2)?

- Gaussian path approximation in (7) looks like a strong assumption. It is not clear from theorem 5 whether it is always possible to decrease L in case we base our computational algorithm on such assumption. Are there any other assumptions such that we can get closed form expressions like (8)?

---

> ### Author Response · Authors · 2023-11-17
> **Author Response to Reviewer 7Jx7 (part 1/2)**
>
> **1. Computational analysis**
>
>
> - Here, we provide further computational analysis in addition to Table 5, which is included in the initial submission, showing that our GSBM algorithm runs efficiently & outperforms prior methods. Specifically, GSBM (Alg 5) can be decomposed into 3 components: _(i)_ the matching algorithm given the probability path $p_t$ (line 1 in Alg 5), _(ii)_ simulating the coupling (line 4), and, finally, _(iii)_ solving the CondSOC (Eq 6) using Gaussian spline (lines 5-6). As the first two components are inherited from prior matching algorithms for SB & OT, they exhibit _identical_ computational complexity as in training or sampling from (denoising) diffusion models.
>
>
> - On the other hand, the third component (_i.e._, Alg 3) is introduced as a result of our alternating optimization scheme in GSBM, and we agree that an expanded computational analysis would be beneficial. Below, we report the runtime and memory complexity of Alg 3 w.r.t. varying sample sizes. These two tables correspond, respectively, to crowd navigation (dimension $d$=2) and image transfer ($d$=3x64x64=12288), serving as examples of low and high dimensionality, as suggested by the reviewer. All values are measured on a V100 32G GPU.
>
>   | sample sizes (2D) | 128 | 256 | 512 | 1024 |
>   |---|---:|---:|---:|---:|
>   | runtime (sec/itr) | 0.01 | 0.01 | 0.01 | 0.01 |
>   | memory (GB) | 0.799 | 0.825 | 0.861 | 0.907 |
>
>   | sample sizes (image) | 8 | 16 | 32 | 64 |
>   |---|---:|---:|---:|---:|
>   | runtime (sec/itr) | 0.01 | 0.03 | 0.06 | 0.06 |
>   | memory (GB) | 2.383 | 4.315 | 6.849 | 11.393 |
>
>
> - For low dimensional problem (first table), the runtime stays constant w.r.t. sample sizes, as the GPU hasn't reached its maximum utility. For high dimensional case (second table), the runtime grows roughly linearly, which is also expected. We emphasize that in both cases, the optimization in Alg 3 exhibits fast convergence, terminated in at most 10 seconds for both 2D and images. This is attributable to the simplified boundary constraints and the significantly fewer parameters in splines compared to, _e.g._, the network that parametrizes the SDEs. Using image transfer as an illustration, the count of spline parameters is only 2-9% in comparison to the UNet.
>
>
> - Regarding memory complexity, both tables indicate linear growth w.r.t. sample sizes. It is noteworthy that, given the rapid convergence of Alg 3, one might contemplate reducing the sample size for a more favorable memory consumption. Such reduction does not impact the solution, as the splines are constructed independently for each pair of $(x_0,x_1)$. For completeness, the table below outlines the memory consumption of other relevant baselines, highlighting that our GSBM scales to much higher dimensions, as also shown in Table 6. All values are measured on a V100 32G GPU in units of GB. Note that NLSB incurs out-of-memory error beyond 100 dimensions.
>
>
>   |  | crowd navigation ($d$=2) | lidar ($d$=3) | opinion ($d$=1000) | AFHQ ($d$=12288) |
>   |---|---|---|---|---|
>   | NLSB [1] | 2.3 | 2.5 | n/a | n/a |
>   | DeepGSB [2] | 2.8 | 3.1 | 22.3 | n/a |
>   | **GSBM (ours)** | 0.97 | 1.1 | 16.2 | 30.1 |
>
> ---
>
> **2. Clarification on image data**
>
>
> - We emphasize that our GSBM algorithm _does_ operate on image data within the full pixel space at resolution 64 (thus the problem dimension is $d$=3x64x64). The latent space appearing in Sec 4.2 served solely as a means to construct a state cost $V(x)$ that takes into account some semantic information for image data. As $V(x)$ is defined as a L2 regression loss between two _images_ (see  Eq 14), the latent space was _never_ used to reduce the problem dimension. We hope the reviewer will recognize this distinction. Our generalized construction allows us to make use of nonlinear state costs, and we chose to showcase this by adapting a latent space. It may also be possible to consider other perceptual distances, such as LPIPS [3]. In the revision, we add clarification in Sec 4.2 (marked dark red above Eq 14) to avoid future confusion, and we thank the reviewer for bringing up the topic.
>
> ---
>
> [1] Neural Lagrangian SB (https://arxiv.org/abs/2204.04853)
> [2] Deep generalized SB (https://arxiv.org/abs/2209.09893)
> [3] Unreasonable effectiveness of deep features as perceptual metric (https://arxiv.org/abs/1801.03924)

---

> ### Author Response · Authors · 2023-11-17
> **Author Response to Reviewer 7Jx7 (part 2/2)**
>
> **3. Strategy for initializing the controlled points in spline optimization (Alg 3)**
>
> - In practice, we find that the spline optimization (Alg 3) is quite robust to hyper-parameters. The two tables below provide a quantitative demonstration of this. We report the optimized objective values (Eq 6a) by initializing Alg 3 with variations in how the controlled points are distributed across time steps (regular (first table) vs irregular (second table) and different numbers of controlled points for parameterizing the mean $\mu_t$ and standard deviation $\gamma_t$ splines (see Eq 9). The initial objective value is much larger relatively, around 18000. Hence, it is evident that across all configurations, the optimization converges to similar values. Furthermore, we observe stable convergence in all cases. We highlight these results as a consequence of how our GSBM decomposes the GSB problem (see Sec 3.1), which yields a variational problem (Prop 2 & Eq 6) that can be optimized more simply and steadily (since it is constrained by two end-points instead of two distributions).
>
>   |  | Num of controlled points in $\mu_t$ = 5 | 10 | 15 | 20 |
>   |---|---|---:|---:|---:|
>   | Num of controlled points in $\gamma_t$ = 10 | 448.71 | 449.80 | 439.11 | 439.92 |
>   | 20 | 454.71 | 452.88 | 439.81 | 439.92 |
>   | 30 | 459.50 | 456.13 | 444.10 | 443.14 |
>   | 40 | 462.24 | 459.23 | 446.05 | 446.34 |
>
>   |  | 5 | 10 | 15 | 20 |
>   |---|---:|---:|---:|---:|
>   | 10 | 449.37 | 449.74 | 439.39 | 440.04 |
>   | 20 | 449.22 | 454.84 | 440.91 | 441.28 |
>   | 30 | 455.23 | 454.64 | 443.15 | 443.64 |
>   | 40 | 464.27 | 454.97 | 446.19 | 445.31 |
>
> ---
>
> **4. Gaussian path approximation**
>
> - This is an interesting question, and we thank the reviewer for raising it. We do acknowledge that Gaussian path approximation to the CondSOC (Eq 6) may be different from the actual minimizer necessary to establish the convergence analysis in Thm 5. Specifically, Thm 5 & 6 assume ideal conditions when Stages 1 & 2 converge to the actual global minimum. Although these ideal conditions may not always be met in our practical instantiations of GSBM, we emphasize that they offer valuable insights that validate our alternating optimization procedure.
>
> - That being said, one might explore alternative solvers for CondSOC that, albeit are less efficient, eliminate the need of Gaussian pat approximation in Stage 2. This can include _(i)_ solving Eq 6 directly with path integral as in our Prop 4, _(ii)_ enforcing the HJB PDE in Eq 18 with an PINN loss [4,5], or _(iii)_ expanding Eq 18 with nonlinear Feynman Kac lemma and then applying temporal difference loss, as initially proposed in [2]. In practice, we find that Gaussian path approximation within GSBM offers much better tradeoffs between efficiency/scalability and optimality, as compared to other deep learning-based approaches.
>
> - With regards to other scenarios in which Eq 6 admits closed-form expressions, we've identified two cases. First, for a sufficiently large $V$ that penalizes deviation from a compact manifold and $\sigma :=0$, Eq 6 amounts to solving the geodesic path, which admits analytic solutions for manifolds such as sphere, tori, hyperbolic ball, and SPD matrices. Secondly, for quadratic $V$ and $\sigma :=0$, Eq 6 reduces to a linear quadratic regulator with two end-point constraints. As we detail in Appendix E (Page 25), its solution relates to a Lyapunov differential equation that admits analytic solutions. For general state costs considered in Sec 3.2, solving Eq 6 necessitates solving its necessary condition in Eq 18 (see Page 16). Hence, as long as Eq 18 has analytic solution for some given $V$, so does Eq 6.
>
> ---
>
> [4] Physics-informed neural networks (https://arxiv.org/abs/1711.10561)
> [5] Transport, variational inference and diffusions (https://arxiv.org/abs/2307.01050v1)

---

> ### Comment · Area_Chair_7Gp2 · 2023-11-20
> **Respond to authors' rebuttal**
>
> Please, confirm that you have read the author's response and the other reviewers' comments and indicate if you are willing to revise your rating.

---

### Author Response · Authors · 2023-11-17
**Author response to all reviewers**

We thank the reviewers for their valuable comments. We are excited that the reviewers identified the importance of the problem (Reviewer 7Jx7, fcZ9), novelty of our technical contributions (all reviewers), acknowledged our extensive experiments (all reviewers), and found the paper well-written (Reviewer 7Jx7, AA2i, fcZ9). We believe our GSBM takes a significant step that advances modern successes of diffusion models & matching algorithms to more general distribution matching problems.

---

As _all_ reviewers recognized our technical novelty, the primary criticisms (raised by Reviewer cBUM) stemmed from the insufficient clarification on the presentation between the GSBM algorithm and GSB problem, and a few missing references and discussions. We agree that these aspects might be unclearly stated in the initial submission, which led to unnecessary confusion and could mislead readers. This was never our intention.

In the revision, we revise Sec 1 carefully and include additional clarifications scattered across abstract, Sec 2, Sec 3, and conclusion. Notable changes are marked in dark red in the revision and enumerated below. Additional results on experiments/derivation are included in Appendix E on Page 25.

- We reiterate that our GSBM algorithm, akin to most modern deep learning-based algorithms for OT/SB, provide approximate solutions to the GSB problem (rather than exact solutions). These changes have been incorporated throughout the paper, with primary revisions made in Sec 1 and fewer adjustments in abstract, Sec 3.2, and conclusion.

- Missing references and discussions such as OT with general costs, feasibility, and matching between SDE and density have been added to Sec 1, 2, and 3.1, for improved rigor.

- The original Table 1 has been moved to Appendix A (now as Table 6) in order to accommodate these changes (kindly note that we're not allowed to add new page to the main paper during rebuttal this year). Furthermore, we present a summary in Fig 9 illustrating the distinctions in optimality structures among various classes of matching algorithms.

- Finally, in Appendix E, we provide additional experiments in Fig 16 demonstrating how distinct couplings emerge from nontrivial $V$ and discuss Lemma 3 in the limit of deterministic dynamics (_i.e._, $\sigma$=0), in which the solution to CondSoc (Eq 6) remains analytic.

We hope that the new presentation is clearer in presenting the contributions of our method in tackling a broader class of distribution matching problems. We try our best to resolve all raised concerns in the individual responses below.

---

### Meta-Review · Area_Chair_7Gp2 · 2023-12-07

**Metareview:**

Summary:

The authors consider so-called generalised Schrödinger Bridge Matching problem: unlike standard SBM, which can be formulated as an optimisation problem of a drift of some diffusion, the corresponding optimisation problem in case of GSBM involved an additional state cost. The authors proposed an iterative algorithm to solve the problem. One of interesting tricks, proposed to increase computational efficiency of the solution, is a spline-based approximation. The authors verified the algorithm for GSBM problem on a number of toy and practical tasks.

Strengths:

- the authors consider an import problem statement, which is a very natural generalisation of the SBM problem
- the proposed approach is theoretically sound
- the authors test their approach on a diverse set of problems
- the text is easy to follow
- The authors exposit their work quite well.
- The experimental section is thorough and highlights the myriad applications of the perspectives presented in the paper.
- The method presented is interesting and falls in line well with a series of other related works, highlighting the benefits of controllable paths.
- The paper presents a method which improves upon previous methods by ensuring guaranteed feasibility and stronger performance in high dimension.
- The method seems to clearly outperform comparable methods in the various application settings.
- The paper has rigorous logic, clear expression and abundant experimental details.
- This work broadens the scope of application of Schrodinger bridge matching (SBM) algorithm and converts the existing optimal transport mapping and SBM methods into the special case of variational formulas.
- The feasibility conditions are met at any time during the entire training process, which is a big difference from the previous methods.
- This work opens up new algorithmic opportunities for training diffusion models with task-specific optimal structures.

Weaknesses:

- The authors did not investigate computational limits of the proposed approach
- it seems that the proposed approach does not work in case of image data, only if we consider some latent space like in sec. 4.2.
- In the introduction, it casts OT and Schrodinger Bridge as edge-cases of their variational problem. However, they do not provide an actual solution to that variational problem.
- The constraint (6b) is actually quite hard to enforce.
- The convergence guarantees obtained are merely local, and do not guarantee convergence to a global optimum.
- As noted in the conclusion, the method requires a differentiable state cost.
- The Generalized Schodinger Bridge Matching (GSBM) proposed in this paper is not applicable to any form of V_{t}, but requires the differentiability of V_{t} and relies on quadratic control costs to establish its convergence analysis.

Recommendation:

All reviewers vote for acceptance. I, therefore, recommend acccepting the paper and encourage the authors to use the feedback provided to improve the paper for the camera ready version.

**Justification For Why Not Higher Score:**

Two reviewers are only weakly leaning toward acceptance.

**Justification For Why Not Lower Score:**

All reviewers vote for acceptance.

---

### Decision · Program_Chairs · 2024-01-16

Accept (poster)